# Bidirectional thermo-regulating hydrogel composite for autonomic thermal homeostasis

Gyeongsuk Park[1,3], Hyunmin Park[1,3], Junyong Seo [2], Jun Chang Yang[1], Min Kim[1], Bong Jae Lee [2] & Steve Park [1] ✉

Thermal homeostasis is an essential physiological function for preserving the optimal state of complex organs within the human body. Inspired by this function, here, we introduce an autonomous thermal homeostatic hydrogel that includes infrared wave reflecting and absorbing materials for improved heat trapping at low temperatures, and a porous structure for enhanced evaporative cooling at high temperatures. Moreover, an optimized auxetic pattern was designed as a heat valve to further amplify heat release at high temperatures. This homeostatic hydrogel provides effective bidirectional thermoregulation with deviations of 5.04 °C ± 0.55 °C and 5.85 °C ± 0.46 °C from the normal body temperature of 36.5 °C, when the external temperatures are 5 °C and 50 °C, respectively. The autonomous thermoregulatory characteristics of our hydrogel may provide a simple solution to people suffering from autonomic nervous system disorders and soft robotics that are susceptible to sudden temperature fluctuations.

Thermal homeostasis is a vital physiological function for the human body to maintain an optimal temperature of 36–37 °C. It prevents medical emergencies such as hyperthermia and hypothermia, which if left untreated can severely damage organs and their constituents[1–4]. The human body retains heat by narrowing the capillary vessels underneath the epidermis to reduce emissive thermal radiation and by decreasing sweat secretion to reduce evaporative cooling. Conversely, heat is released through the widening of capillary vessels and by increasing sweat secretion[5,6]. Mimicking such properties is of great scientific interest and has a wide variety of applications, such as restoring body temperature control for patients with impaired autonomic nervous systems or facilitating the operation of soft robotic systems that are sensitive to temperature fluctuations.

One approach to mimicking thermal homeostasis has been to develop functional fabrics with cooling or heating capabilities (Supplementary Table 1). Porous and hierarchical nano- to micro-fibrous structures that can regulate infrared radiation (IR) for radiative cooling were reported[7–11]. Other fabrics can impart a heating effect either by

using a conductive material to enhance the reflectivity of thermal radiation or by using joule-heating devices that generate thermal energy[12–16]. However, such devices are only capable of unidirectional temperature control (i.e., either cooling or heating). The use of a thermoelectric device enables bidirectional temperature control (i.e., both cooling and heating)[17–19]. However, these devices require multiple fabrication steps, complicated routing and wiring on a circuit board, and an external power source, which render them undesirable for wearable applications that need a simple design. Furthermore, the development of an autonomic device that can control the temperature without conscious input from the user is still a major challenge.

Stimuli-responsive hydrogels such as poly(N-iso-propylacrylamide) (PNIPAm) are promising materials for autonomic temperature control because of their phase transition–assisted actuating properties without needing external energy[20–23]. PNIPAm expels water from its hydrogel network when the environmental conditions exceed the lower critical solution temperature (LCST), which has facilitated research into diverse applications such as

[1]Department of Materials Science and Engineering, Korea Advanced Institute of Science and Technology, Daejeon 34141, Republic of Korea. [2]Department of Mechanical Engineering, Korea Advanced Institute of Science and Technology, Daejeon 34141, Republic of Korea. [3]These authors contributed equally: Gyeongsuk Park, Hyunmin Park. ✉e-mail: stevepark@kaist.ac.kr

sweating roofs[24], pharmaceutical packaging[25,26], thermal switching film[27], and sweating actuators[28]. However, such studies did not take full advantage of the inherent flexibility of PNIPAm, which enables conformal attachment on human skin. Although a skin-applicable heat valve was introduced to realize artificial perspiration[29], heat trapping below LCST was not demonstrated, and the closed/open areal ratio of the valve was not fully optimized. To realize a thermoregulatory system applicable to the human skin, a hydrogel with bidirectional temperature control is needed. However, this is difficult due to the intrinsic high IR transmittance of PNIPAm.

In this work, we introduce an autonomous thermal homeostatic hydrogel (ATHH) that is attachable to the skin and is capable of reversible and bidirectional thermal control (Fig. 1a, b). Firstly, ATHH consisted of uniformly dispersed nano- to micro-size conductive composite to achieve a low IR transmittance of less than 1%, which allows it to act as a thermal insulator under low surrounding temperature. Secondly, the porous structure increases the evaporation rate, and the optimal actuating design generates high closed/open areal ratio at the adjusted LCST of 35.7 °C (which is similar to that of human body temperature. See Supplementary Fig. 1), accelerating the heat release under high surrounding temperature. Overall, such attributes of ATHH enables effective maintenance of human body temperature, rendering it highly promising for various bidirectional autonomous thermal regulating e-skin applications.

## Results

### Structural effect and material properties of ATHH

To trap heat at low surrounding temperature, a composite material capable of reflecting or absorbing IR wave was embedded in ATHH. ATHH contained nano- to micro-size silver particles for reflecting IR and a conducting polymer, polypyrrole (PPy), possessing mid-gap states created by polarons and bi-polarons, for absorbing IR wave[30-33]. To fabricate ATHH, we first dissolved silver nanowires (AgNWs), which have an inherently high IR reflection, in a NIPAm hydrogel solution. The AgNWs were pulverized into micron-size particles during NIPAm polymerization involving UV irradiation (Supplementary Fig. 2)[34]. Next, the AgNW/PNIPAm hydrogel was immersed in a solution of pyrrole monomers and silver nitrate (used as the oxidant) for pyrrole polymerization within the hydrogel. This process generated a porous structure by significantly reducing the particle size from 1 μm to 400 nm[35,36]. We confirmed that the porous structure was formed due to the size reduction effect of PNIPAm particles composited with pulverized AgNW frame during polypyrrole synthesis (Supplementary Fig. 3). The reduction of silver cations from the oxidant formed additional spherical silver particles at the surface and inside of the hydrogel (Supplementary Fig. 4). The synthetic methods, size of particles and composition of reagents were determined by experimental analyses as shown in Supplementary Figs. 5–9.

PNIPAm has a high transmittance at a wavelength range of 3–14 μm, which is associated with body heat. Thus, on its own, this material lets out most of the thermal radiation emitted by the human body, leading to considerable heat loss (Fig. 2a). PNIPAm with decomposed AgNW (i.e., AgNW/PNIPAm) has reduced transmittance compared to that of the pure PNIPAm. After the polymerization of polypyrrole (ATHH), transmittance is significantly reduced with an average of less than 1%, which can be attributed to the enhanced IR reflection due to additional Ag particles and IR absorption by polypyrrole (Supplementary Fig. 10). Other conducting polymers could also be embedded into PNIPAm to considerably lower the transmittance (Supplementary Fig. 11). Such characteristics of ATHH were verified empirically by visual inspection of images taken by an IR camera. Dried ATHH and PNIPAm loaded on skin showed deviations of 6 °C and 3 °C, respectively, with respect to the surface temperature of the bare forearm after the saturation temperature was reached (Fig. 2b

and Supplementary Table 2). These results confirm that ATHH successfully blocked IR from the human body.

We measured the normal emissivity spectra $\varepsilon_\lambda$ of PNIPAm and ATHH when mounted on a skin model made of Ecoflex (which has a thermal conductivity and emissivity similar to those of the epidermis) and calculated their respective emissive heat fluxes $E(T)$ (Supplementary Note 1). PNIPAm was calculated to have $E(T)$ of 138–261 W m$^{-2}$ at 10 °C–50 °C, while ATHH had $E(T)$ of 119–223 W m$^{-2}$ in the same temperature range (Fig. 2c). In other words, ATHH reduced the total heat loss by 17% compared with PNIPAm by blocking heat emissions, furthermore verifying ATHH's superior thermal insulating capability compared to that of PNIPAm.

To specifically analyze the heat trapping effect of the nano- to micro-size Ag particles, we first used scanning electron microscopy to examine the morphology and distribution of the Ag particles. Ag particles on the top surface of ATHH were uniformly dispersed with sizes varying between 50 nm and 2 μm (Fig. 2d). The detected surface particles were verified as being silver by energy-dispersive spectroscopy (EDS) and X-ray diffraction analysis (Supplementary Figs. 12 and 13). For the ATHH, a highly porous structure was observed in the cross-sectional image (Fig. 2e). Unlike PNIPAm and AgNW/PNIPAm, ATHH shows a highly porous structure regardless of the drying method due to the reduction in the hydrogel particle size through the synthesis of polypyrrole hydrogels and an interference effect of AgNW on polymer chain conformation (Supplementary Figs. 14–16). Utilizing transmission electron microscopy for bright-field image analysis, nano- to micro-size spherical and rod-shaped particles were found inside of the ATHH (Fig. 2f). The particles were confirmed as being silver through EDS using high-angle annular dark-field scanning transmission electron microscopy (HAADF-STEM) (Supplementary Fig. 17). These results confirmed that silver particles with diverse sizes were on the surface and embedded in the porous structure of ATHH, which can be attributed to the reduction of silver cations and pulverized AgNWs.

To determine the effect the shape and size of the silver particles have on the radiative heat flux, scattering effect of the particles caused by the electromagnetic interaction between the silver particles and incident light (Supplementary Note 2 and Supplementary Fig. 18) was analyzed. The scattering efficiency of rod-shaped and spherical silver particles (Fig. 2g) were calculated, and for both particles, the efficiency was affected by the particle size. The micro-size silver rods had a much higher scattering efficiency than the micro-size spherical particles owing to their differing excitation modes[37], and the rod-shaped silver particles with a length of 1–2 μm had a high scattering efficiency in the target IR region. Overall, both the micro-size silver rods and spheres enhanced the reflectivity of ATHH due to their strong volumetric scattering effect (Fig. 2h), thus enhancing heat trapping performance of ATHH compared with PNIPAm. We also conducted the effect of porosity and pore sizes utilizing computational simulation (Supplementary Note 3 and Supplementary Fig. 19).

When the surrounding temperature is high, our body secretes sweat to cause evaporative cooling, thereby lowering the body temperature. To mimic this cooling effect, it is important to enhance the evaporative cooling capability of the hydrogel. In this regard, the open networked pores of ATHH is advantageous as they form microfluidic channels that promote internal fluid circulation and evaporation via the capillary effect[8]. The pore size distribution of the ATHH cross-section indicates the formation of micron-size pores (Fig. 2i). The ATHH exhibited a higher evaporation rate by 130% compared with PNIPAm at room temperature (i.e., below the LCST) and 40 °C (i.e., above the LCST), thus demonstrating its superior evaporative cooling capability (Fig. 2j).

### Characteristics of heat valve-based ATHH

Although ATHH possesses a superior heat trapping at low surrounding temperature, for bidirectional thermoregulation, it is

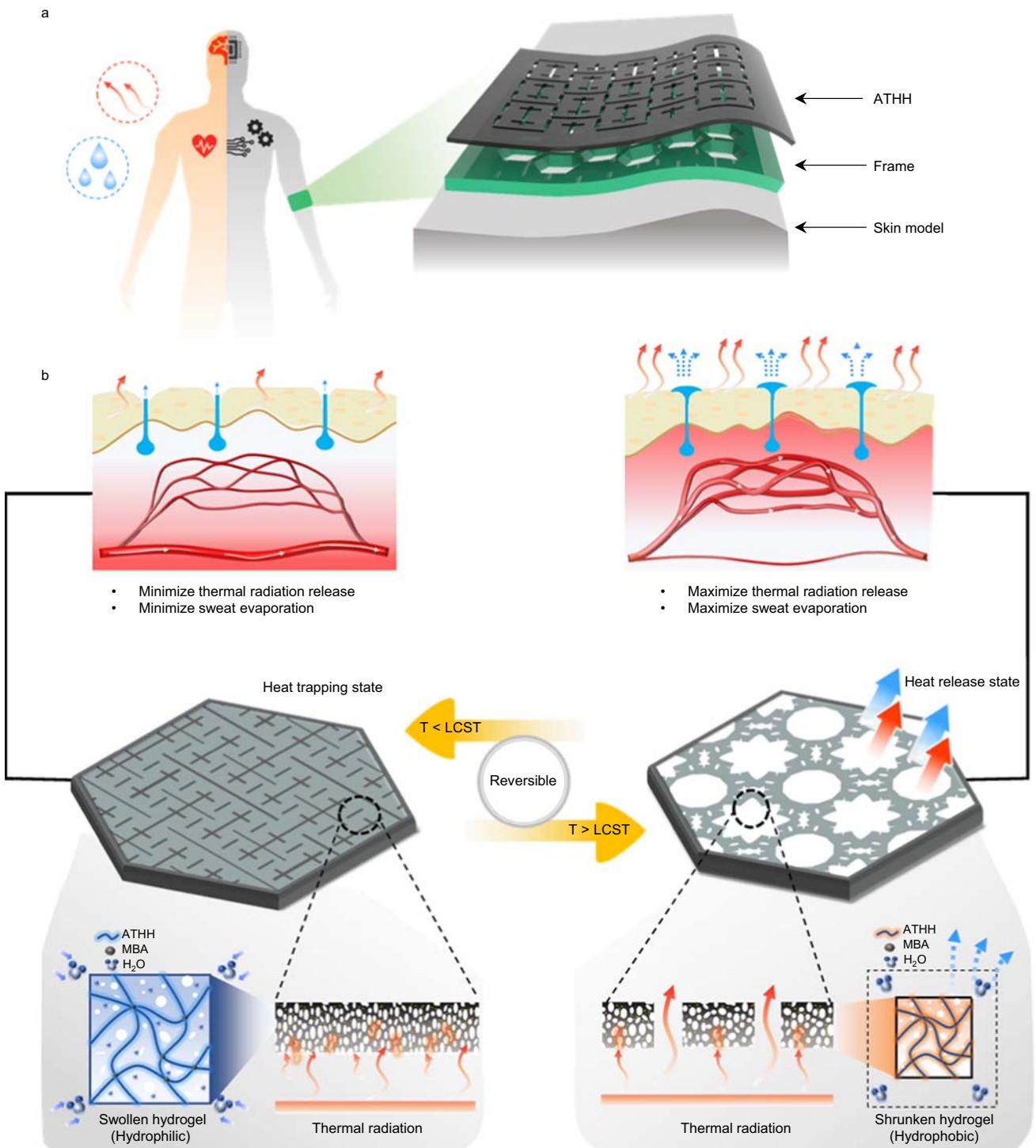

**Fig. 1 | Human inspired design of an autonomous thermal homeostatic hydrogel (ATHH). a** Schematics of thermal homeostasis of the human and ATHH device. **b** Thermo-responsive hydrogel based thermal homeostasis of ATHH analogous to the autonomic nervous system of the human. Heat trapping state due to the low transmittance of infrared radiation (IR) below lower critical solution temperature (LCST) (left). Heat release state due to the unclosed auxetic patterns maximizing IR release and enhanced evaporative cooling attributed to highly porous structures with microfluidic channels at hydrophobic state over LCST (right).

important to promptly switch to heat release state as the surrounding temperature rises. In this regard, hydrogels can be patterned to create a valve that opens and closes as phase transition occurs. For this study, auxetic fractal pattern[38] was utilized due to its negative Poisson's ratio that would render the hydrogel mechanically stable in the stretched state[39] and to maximize the closed/open areal ratio. As depicted in Fig. 3a, pattern level 3 was determined to be the optimal,

with a closed/open areal ratio of 7.3, along with mechanical stability (Supplementary Fig. 20). Our pattern was programmed to realize a maximum stretchability of 108% from the original state (Fig. 3b), which enabled the opened area to be 70% of the total area (Fig. 3c and Supplementary Movie 1) above the LCST. Rapid response and temperature sensitivity of ATHH were demonstrated in Supplementary Fig. 21.

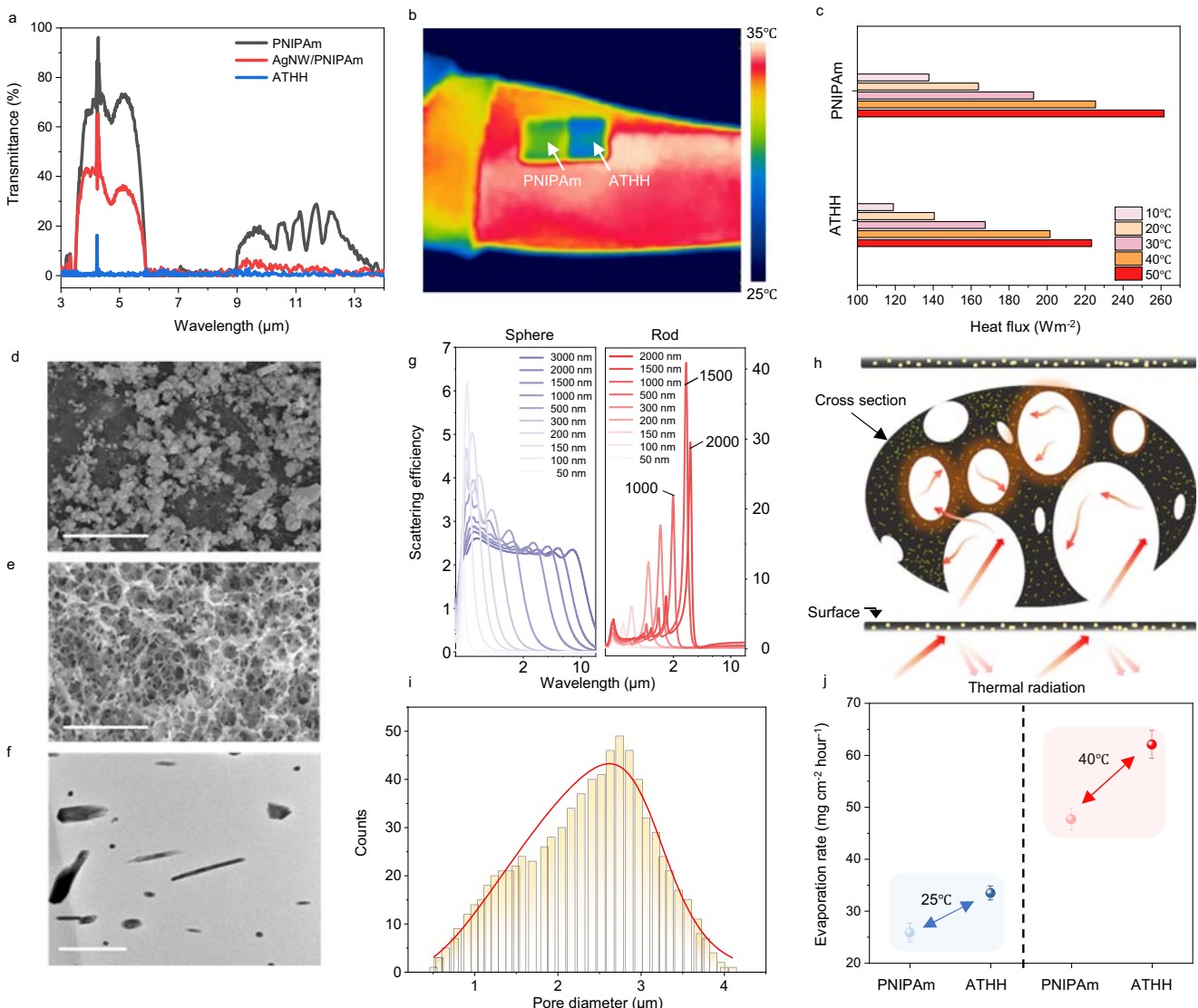

**Fig. 2 | Structural effect and material properties of ATHH. a** The Infrared transmittance spectra of hydrogels composed of respective materials. **b** IR camera image of PNIPAm and ATHH on a human skin. **c** Calculated heat flux from the IR camera image of **b**. **d**–**e** Top and cross-section SEM images of ATHH, respectively. Scale bars, 10 μm. **f** TEM image of nanomaterials in ATHH. Scale bar, 1 μm. **g** Scattering efficiency of nano to micro particles which hinders propagation of thermal radiation, leading to the IR scattering of ATHH along with the particle size. **h** Schematic of thermal scattering and heat trapping. **i** Pore size distribution of ATHH. **j** Evaporation rate of PNIPAm and ATHH above and below LCST of hydrogels. Error bars indicate standard deviation from the mean.

IR transmittance of the patterned ATHH was measured when it was opened and closed, (Supplementary Fig. 22 and Fig. 3d). The IR transmittance was 55.1% with an open pattern but was dramatically reduced to 3.2% with a closed pattern. Figure 3e is a calculation of insulated heat (i.e., trapped heat due to the mounting of hydrogel film on the skin model) using dried samples (to exclude effect of water evaporation) during temperature transition from 25 °C to 45 °C (See Supplementary Note 4). In the case of non-patterned samples of PNIPAm and ATHH, insulated heat continuously increases even as temperature increases, signifying that these samples do not have cooling capability. As expected, ATHH insulates more heat due to lower IR transmittance. For the patterned samples, at the LCST, the insulated heat decreases due to heat release resulting from the opening of the patterns. Patterned ATHH undergoes a large change in insulated heat (26.01 W m$^{-2}$ at 25 °C and only 11.44 W m$^{-2}$ at 45 °C) with an LCST (35.7 °C) similar to that of the body temperature. In terms of the difference in insulated heat from minimum to maximum value, patterned PNIPAm and ATHH had values of 5.71 and

16.06 W m$^{-2}$, respectively, confirming improved heat regulation of ATHH by a factor of 2.8.

To quantitatively analyze the evaporative cooling effect, the effective evaporative area ratio (EEAR) was calculated as a function of temperature (Fig. 3f). EEAR is defined as the value of evaporative area of the hydrogel after opening the auxetic pattern divided by original evaporative area before opening. EEAR gradually decreases with increasing temperature because the hydrogel gradually shrinks (Supplementary Fig. 23). Since ATHH has a higher LCST, it has a larger hydrogel evaporative area over the measured temperature range compared to that of PNIPAm (i.e., the additional evaporative area is indicated by the blue colored region). Such additional area indicates that ATHH provides more evaporative area for water in the hydrogel to absorb heat and be evaporated, thus resulting in more effective cooling. Figure 3f also depicts evaporative cooling power as a function of temperature (evaporation rate multiplied by latent heat of vaporization). PNIPAm featured a similar evaporation rate to that of previously reported[28]. At 45 °C, the cooling power of PNIPAm

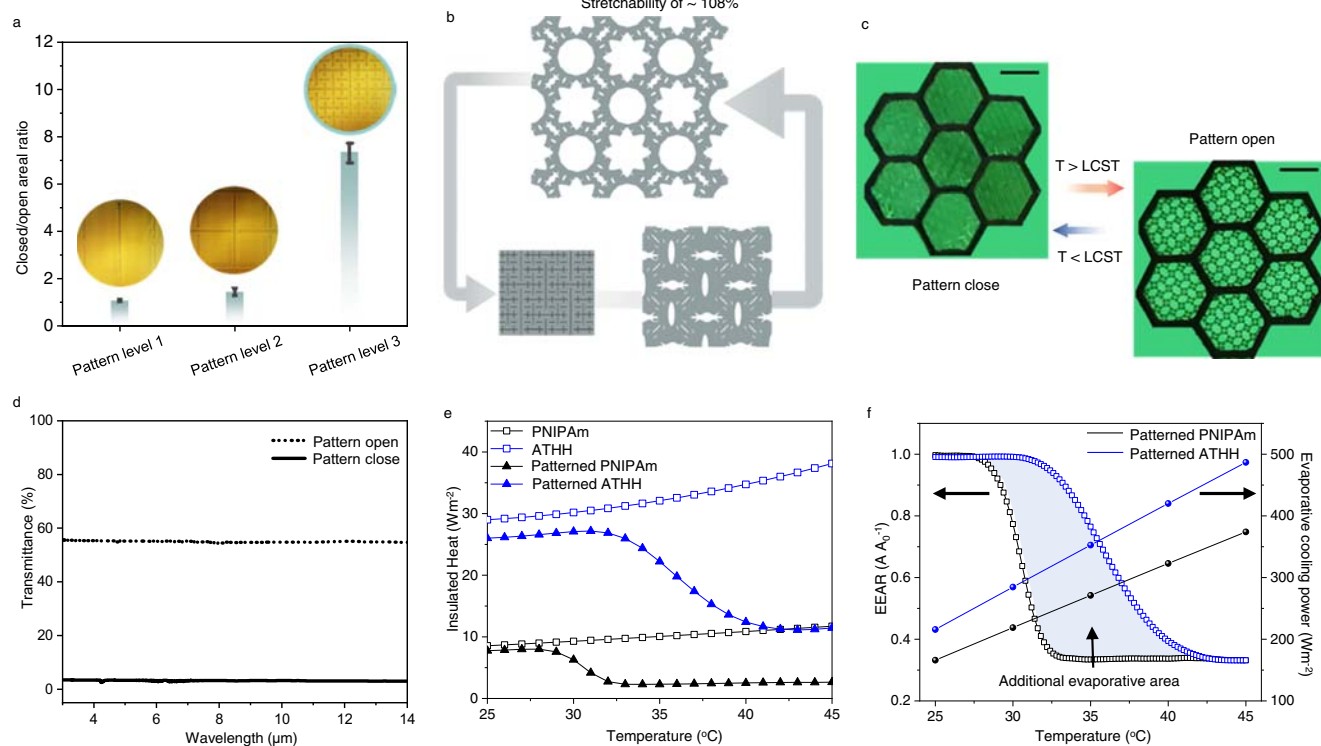

**Fig. 3 | Characteristics of heat valve-based ATHH. a** Closed/open areal ratio of each auxetic pattern applied to the ATHH. Error bars indicate standard deviation from the mean. **b** Schematics of auxetic-patterned heat valve in the ATHH. **c** Photographs of reversibly closing and opening ATHH on the fluorescence paper according to the LCST. Scale bars, 1 mm. **d** Pattern adapted transmittance within Infrared radiation range. **e** An insulated heat during the temperature transition from 25 °C to 40 °C. **f** Effective evaporative areal ratio (EEAR) and evaporative cooling power of patterned PNIPAm and ATHH.

was 374 W m$^{-2}$ while that of the ATHH was 487 W m$^{-2}$. As mentioned above, this can be attributed to the porosity of the ATHH. These results together further verify that ATHH effectively releases heat in the cooling state. Furthermore, practical time available for ATHH usage is regulatable by its thickness, and ATHH can be rapidly restored within 2 min without damage and distortion through water supply even after being dehydrated (Supplementary Figs. 24 and 25).

## Autonomic thermoregulation by ATHH

To compare the real-time thermoregulation performances of the patterned hydrogels, a temperature sensor was inserted into the center of the skin model to monitor the temperature changes via a microcontroller unit. The patterned ATHH was attached on top of a supporting substrate (made with Polyacrylamide (PAAm)/Alginate) with large openings to ensure stable mechanical motion when attached on the skin model (see Supplementary Fig. 26 and methods for fabrication process, and Supplementary Fig. 27 for large-scale processability). These devices will be referred to as S-PNIPAm and S-ATHH hereafter, and the humidity for thermoregulation is maintained at 40%. In the first test, S-PNIPAm and S-ATHH were each place on skin models (which were all at room temperature), and was subsequently placed on a hot stage fixed at 40 °C (Fig. 4a). Aluminum sample was used as a reference, which is widely used for heat trapping applications because it has an IR reflectance of ~99%. We compared the temperatures of the skin models when the aluminum film-covered skin model reached 36.5 °C (Fig. 4b, top). The S-ATHH, S-PNIPAm, and bare skin models reached temperatures of 35.56 °C, 33.38 °C, and 33.06 °C, respectively, which differed from their initial temperature by 6.56 °C, 4.38 °C, and 4.06 °C, respectively (Fig. 4b, bottom). In terms of the time taken to reach 36.5 °C, the S-ATHH skin model required an additional 39 s compared to that of the aluminum film. In contrast, the S-PNIPAm and bare skin models required additional times of 196 and 224 s, respectively. This demonstrates that S-ATHH traps heat more effectively than S-PNIPAm and bare skin models.

To compare the cooling effects, we firstly saturated the temperatures of the skin models at 40 °C by placing them on a hot stage. Next, we placed the S-ATHH and S-PNIPAm on top of the skin models and tracked temperature in real time for 600 s (Fig. 4c). The S-ATHH skin model had a lower minimum temperature with a deviation of 5.87 °C from the initial temperature of 40 °C at 150 s, while the S-PNIPAm model had a deviation of 3.75 °C (Fig. 4d). This demonstrates that S-ATHH was more effective at cooling, which as mentioned above, we attributed to the microfluidic channels of the porous structure and additional evaporative area as shown in Fig. 3f. After 600 s, the S-ATHH model maintained a temperature of 35.25 °C, demonstrating the sustainability of the cooling effect.

To further confirm the bidirectional temperature controllability, S-ATHH and S-PNIPAm placed on skin models were mounted on a hot plate that maintains a temperature of 36.5 °C. This set-up was then placed inside of an environmental temperature control unit. The external environmental temperature was set to gradually increase from 5 to 50 °C (Fig. 4e). The S-ATHH enabled the skin model to maintain a temperature closest to the homeostasis criterion at both low and high environmental temperatures (Fig. 4f), thus verifying its ability for thermal homeostasis over a large temperature variation. The effect of thickness was revealed the thicker S-ATHH reinforced heat trapping at low temperature, however, this also diminished cooling ability by interrupting heat release at high temperature (Supplementary Fig. 28). The reliability and repeatability of S-ATHH was demonstrated through cyclic heating and cooling with the temperature deviation of 5.04 °C ± 0.55 °C and 5.85 °C ± 0.46 °C from 36.5 °C, when the external temperatures were 5 °C and 50 °C, respectively (Supplementary Fig. 29).

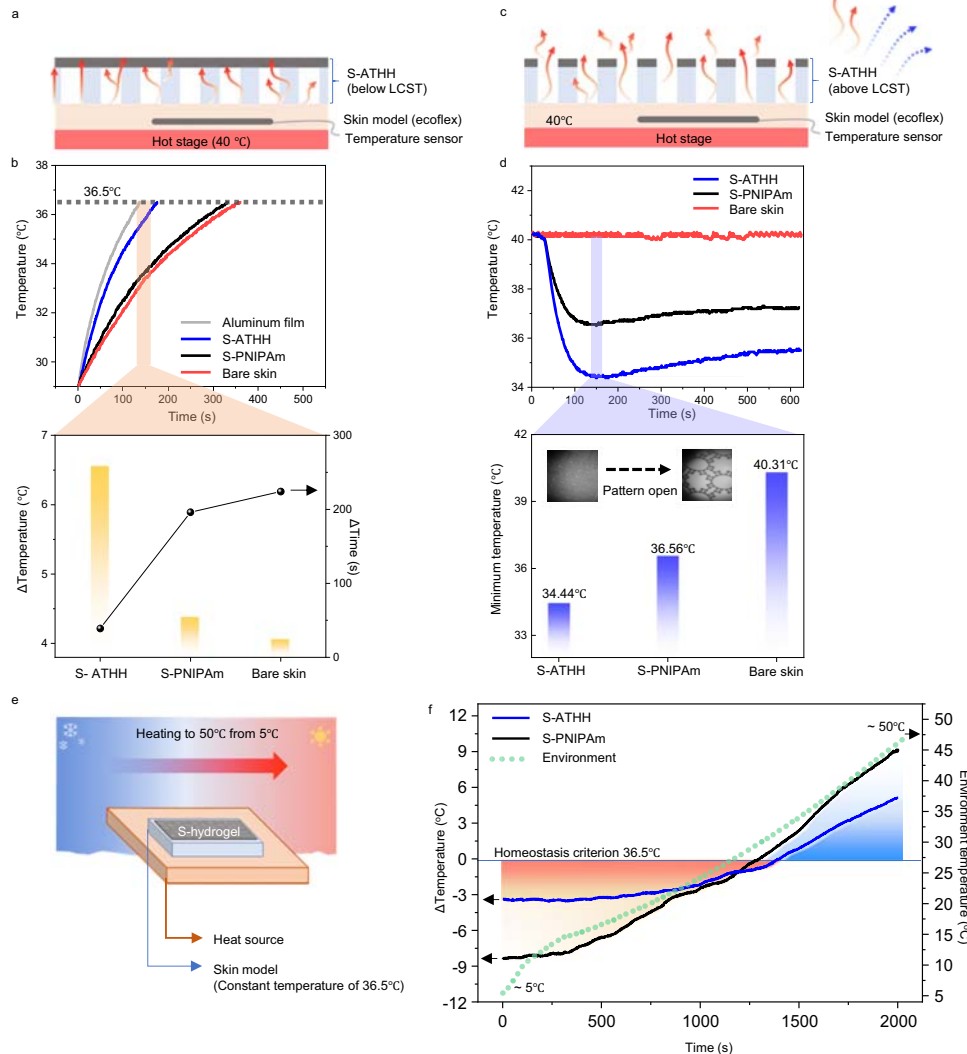

**Fig. 4 | Autonomic thermoregulatory abilities of ATHH for thermal homeostasis. a** Schematic of experimental setups for heating effect. **b** Heating ability comparison, reaching target temperature until 36.5 °C (top). Temperature difference from an initial state and the time consumption to reach 36.5 °C (bottom). **c** Schematic of experimental setups for cooling effect. **d** Cooling ability comparison, placing samples on the hot stage of 40 °C (top), and a minimum temperature

of each samples (bottom). **e** Schematic of measurement setup when the skin models were conserved at an optimum state (36.5 °C), heating environmental temperature from 5 to 50 °C. **f** Real-time temperature deviation measurement from 36.5 °C. Dot line of graph is the environmental temperature of the thermohygrostat.

## Practical demonstrations and applications

We have confirmed the thermoregulatory performance of the ATHH under real-life settings of heating and cooling. Skin models with and without S-ATHH were placed on top of pulp paper (IR transparent) that was suspended in air using a raised platform. IR camera was placed underneath the pulp paper for temperature imaging (Fig. 5a, b). To test cooling effect, steady thermal radiation was generated from a sun-like IR source to reach a temperature of ~60 °C. To test heating effect, skin models were placed outdoors at a low temperature of 10 °C and humidity of 40%. In both cases, the initial temperature of the skin model was 36.5 °C. In the case of cooling, the bare skin model had a yellowish-orange color after 180 s and was indistinct from the surroundings. In contrast, the skin model with S-ATHH was vividly distinct from the surroundings even after 180 s, with a temperature of 40.8 °C, which was a small deviation from the initial temperature (Fig. 5c, d). The inset of Fig. 5c reveals that considerable IR released outward from the opened patterns. In the case of heating, the IR images visually show that S-ATHH skin model retain heat better than that of bare skin model

(Fig. 5e). After 180 s, the skin model with S-ATHH had a temperature of 22.9 °C while the bare skin model had a temperature of 17.2 °C (Fig. 5f).

Stretchability and flexibility are crucial characteristics for skin-attachable application. Therefore, we tested mechanical strain test and conformal attachment on the skin. It was verified S-ATHH could endure up to 30% elongation. S-ATHH also presented conformal attachment even on a bent surface (Supplementary Figs. 30 and 31).

Utilizing temperature-sensitive character of ATHH, we tracked the changes in electrical resistance according to thermal stimulus to evaluate the potential applicability of ATHH towards temperature sensing electronic skin and soft robotics. Thermal energy was provided by repeatedly turning an IR lamp on and off and controlling how long the lamp was on (see methods for details). Reproducible relative changes in resistance of 0.31%, 0.57%, and 0.99% were detected without electrical hysteresis in accordance with thermal stimuli of 7.56, 14.02, and 22.65 J m$^{-2}$, respectively (Fig. 5g). The resistance measurement is possible due to the sufficient conductivity provided by embedded conductive fillers. These characteristics demonstrate that

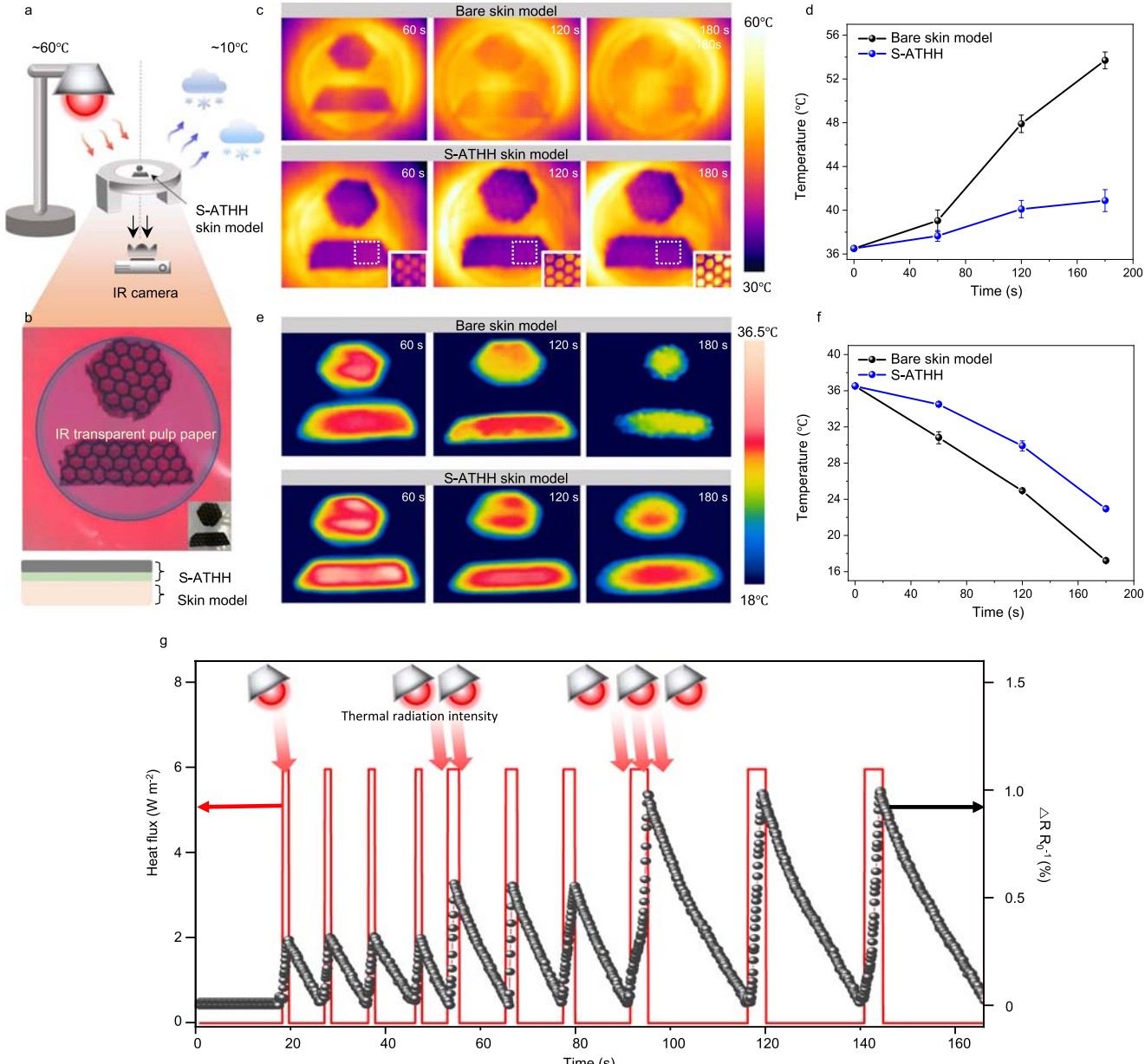

**Fig. 5 | Analysis of ATHH adapted IR Image and its potential application.**
**a** Depiction of real-life measurement for cooling and heating. **b** Photographs of
S-ATHH during IR exposure. S-ATHH mounted on the skin model (Inset).
**c** Sequential IR images of bare and S-ATHH skin model under steady IR exposure for
cooling. IR image of outgoing thermal radiation from ATHH (bottom, Inset).
**d** Temperature measurement of bare and S-ATHH skin models in the heat release
needed state. Error bars indicate standard deviation from the mean. **e** Sequential IR
images of bare and S-ATHH skin models at the chilly outdoors. **f** Temperature
measurement of bare and ATHH skin model in the heat conservation needed state.
Error bars indicate standard deviation from the mean. **g** Relative change in the
electrical resistance according to the thermal stimulus.

ATHH artificial skin can potentially quantify the heat exposure through
changes in resistance.

## Discussion

Stimuli-responsive hydrogels such as PNIPAm are promising mate-
rials to realize autonomously actuatable valves without external
energy. To fully exploit the advantages of PNIPAm for thermal
regulation, additional properties such as heat trapping, enhanced
evaporative cooling, and an LCST close to the standard body tem-
perature is critical. These issues were addressed through the fab-
rication of ATHH, thus enabling bidirectional autonomous thermal
homeostasis. Firstly, silver-polypyrrole composite was uniformly
dispersed through PNIPAm, trapping heat with an average IR
transmittance of less than 1%. An auxetic fractal pattern was uti-
lized

with a high closed/open areal ratio of 7.3, acting as an effective heat
valve. Moreover, the highly porous structure was determined to
enhance evaporative cooling. Experimental results showed that
ATHH offers an enhanced thermoregulatory performance com-
pared to pure PNIPAm. We furthermore confirmed that ATHH can
deliver electrical signals that reflect changes in temperature. The
dehydration of ATHH can limit its long-term usability if there is no
external water supply. However, since increasing the thickness of
ATHH can adjust the dehydration rate and ATHH has considerable
restoration ability of within 2 min even after being fully dehydrated,
we considered this issue resolvable through additional research. We
project that ATHH will be applicable in various fields, such as
treating nervous system disorders and in implementing autono-
mous thermoregulation in soft robotics in the near future.

## Methods

### Materials

All chemicals and solvents were of reagent grade and were used as received without further purification. N-Isopropylacrylamide [$H_2C=CHCONHCH(CH_3)_2$, 99%], acrylamide ($CH_2=CHCONH_2$, 99%), N,N'-Methylenebisacrylamide [$(H_2C=CHCONH)_2CH_2$, 99%] dimethyl sulfoxide (DMSO), 2-Hydroxy-4'-(2-hydroxyethoxy)−2-methylpropiophenone [$HOCH_2CH_2OC_6H_4COC(CH_3)_2OH$, 98%] and 2-Hydroxy-2-methylpropiophenone [$C_6H_5COC(CH_3)_2OH$, 97%] as the photo-initiators, polypyrrole ($C_4H_5N$, 98%), $AgNO_3$ (99%), poly(3,4-ethylenedioxythiophene):poly(styrenesulfonate) (PEDOT:PSS) (1.3 wt% dispersion in $H_2O$), ammonium persulfate [$(NH_4)_2S_2O_8$, 98%], and aniline ($C_6H_5NH_2$, 99%) were purchased from Sigma-Aldrich. AgNW (1.6 wt% in DI water-based solution) was purchased from DS Hi-Metal.

### Fabrication of ATHH and artificial skin

First, 2.14 g of NIPAM monomer (N-Isopropylacrylamide) and 0.12 g of MBA (N,N'-Methylenebisacrylamide) were thoroughly mixed with 1 ml of AgNW solution in the 3 ml of DMSO. The mixed solution was spread over UV Ozone treated glass, and auxetic patterned PDMS stamp through soft-lithography processes pressed the solution without diminutive bubbles which prevent creation of precise auxetic pattern in the mixed solution. Next, pressed solution was cured via UV exposure lamp (365 nm, Liim Tech, KOREA). The substrate consisted of alginate-contained polyacrylamide (i.e., PAAm/Alginate) hydrogel, was also cured on the honeycomb-shaped alumina mold. Both of auxetic patterned PNIPAm and PAAm/Alginate were merged together in the co-polymerization process while free radicals were remained within hydrogel networks using UV curing method. Finally, 40 ml of $AgNO_3$ and 20 ml of polypyrrole solutions were poured over combined hydrogel, and synthesis process was kept for 2 days at the room temperature to synthesize silver particles and polypyrrole. Among conducting polymers, polythiophene cannot be synthesized into PNIPAm networks and, it is difficult to replace pyrrole due to the toxicity of residual reagents. However, polyaniline was readily synthesized by mixing an ammonium persulfate solution (5 wt% of NIPAm) into a solution of 10 ml of 0.1 M HCl and 1 ml ANI in 30 ml of DI water and curing for 1 day. To synthesize PEDOT:PSS embedded AgNW/PNIPAm film, 1 ml of PEDOT:PSS was mixed with already-made AgNW/PNIPAm solution. The solution was cured by UV irradiation for 1.6 seconds. The phase of the silver particles in the hydrogel composite film were verified by X-ray diffractometry (Ultima IV, Rigaku) with Cu Kα radiation. ATHH was completed after thorough washing with DI water for eliminating residues and debris of polypyrrole synthesis. As the artificial skin model, ecoflex (00-20, SMOOTH-ON) was mixed with volume ratio of 1A: 1B, and the mixed solution were degassed for 1 h.

### Real-time temperature deviation measurement

In order to track thermal homeostasis with heating and cooling effect, 36.5 °C fixed PNIPAm and ATHH were placed in the thermo-hygrostat (U1TECH, KOREA). Samples were mounted on the hot plate for steady heat source during experiment duration. Humidity was fixed at 40%, and surrounding temperature was varied from 10 to 50 °C. The temperature sensor inserted into the artificial skin model tracked the real-time temperature fluctuation of PNIPAm and ATHH through an arduino microcontroller.

### Electrical signal response to the thermal stimulus

For tracking electrical signal change, ATHH was combined with silver mesh as the electrode which interconnected ATHH and LCR meter (4284A, HP). The thermal energy of 7.56, 14.02, and 22.65 J m$^{-2}$ applied to the ATHH from IR lamp (150 W, PAR38E, PHILIPS) was shifted at the constant distance for repeatable stimulus as the illuminating time increased. Retaining persistent cooling for the rapid recovery of

temperature to the original state, iced water was located under the ATHH artificial skin. In turn, total resistance of the ATHH was monitored, and the response was defined as a ratio of the resistance change ($\Delta R$) to the resistance in the air ($R_0$).

### SEM, TEM, and specimen preparation

The morphology of the composite materials was observed using SU 8230 field emission SEM (Hitachi). All samples were dried at 90 °C for 3 h in air. For imaging of the surface, the samples were not modified. The cross-sectional images were characterized using cross-cut samples. All samples were deposited ~3 nm osmium film using osmium coater to remove surface charging. TEM and STEM specimens for the hydrogel composite were prepared by lift out via ion-beam milling in a focused ion-beam system (Helios Nanolab 450 F1, FEI). Thin platinum layers were applied over the region of interest before milling to prevent bending of the samples. TEM and HAADF-STEM images were acquired using a transmission electron microscope (Talos F200X, FEI) at 200 kV.

### FT-IR spectrometer measurement

In this study, the normal-hemispherical transmittance ($\tau_\lambda$) and the normal-hemispherical reflectance spectrum ($\rho_\lambda$) of samples were measured by an FT-IR spectrometer (ABB Bomem, FTLA 2000 series) with an integrating sphere. For the transmittance in Fig. 2a, free-standing devices (i.e., pure PNIPAm, AgNW/PNIPAm, and ATHH only) were measured to investigate the opacity characteristics of each device. On the other hand, reflectance spectra of devices loaded on the skin model were measured for estimating the normal emissivity ($\varepsilon_\lambda$) of the system (i.e., hydrogel + skin model). Note that the skin model composed of ecoflex is nearly opaque with the transmittance value less than 1% in the wavelength range of interest (i.e., 7.5–14.0 μm). Therefore, its normal emissivity can be simply obtained from Kirchhoff's law and energy conservation; that is, $\varepsilon_\lambda = 1 - \rho_\lambda$, where $\rho_\lambda$ is the normal-hemispherical reflectance of the system. The normal emissivity spectrum was used to calculate an emitted heat and insulated heat in Fig. 2c. All of the samples were prepared at the dried condition to maintain their transformed shape by phase transition, and we corrected transmittance using real pattern images collected from an optical microscope.

### Experiments on human subjects

All subjects voluntarily involved in experiments after informed consent. The experiments were conducted on a tagaderm-attached human skin to protect physical contact with hydrogel and to regulate additional variables.

## Data availability

The authors declare that the data supporting the findings of this study are available within the article and its Supplementary Information files. Source data are available with the publication or can be requested from the corresponding author. Source data are provided with this paper.

## Code availability

The software code used for rheological modeling of ink is available from the corresponding author upon request.

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

## Acknowledgements

This work was supported by the National Research Foundation of Korea grant funded by the Korea Government (MSIT) (NRF-2022R1A2C2006076 and 2022M3H4A1A03085346) and KAIST UP Program.

## Author contributions

G.P., H.P., and S.P. conceived the concept and designed experiments. G.P. and H.P. conducted the experimental work, including real-time temperature deviation measurement, tracking electrical signal response measurement, and data analysis. J.S. conducted a computational simulation regarding the nano- to micro-sized silver particles, FT-IR measurement using integrated sphere, and emissive heat flux calculation. J.Y. conducted a numerical simulation about mechanical stability of auxetic patterns. M.K. conducted hydrogel composite synthesis. B.J.L. revised computational work and heat flux analysis. S.P. were responsible for managing all aspects of this project. G.P. and H.P. wrote the draft. S.P. revised the manuscript. All authors discussed the results and the manuscript

## Competing interests

The authors declare no competing interests.
