## [Peer Review File · Nature Communications]

Bidirectional Thermo-regulating Hydrogel Composite for Autonomic Thermal HomeostasisReviewers' Comments:

Reviewer #1:

Remarks to the Author:

In this work, the authors prepared a PNIPAM hydrogel incorporated with Ag nano-/microparticles and PPy particles, which showed the ability of bidirectional thermal regulation due to its low IR transmittance at low temperature and the accelerated heat release of porous structure at high temperature. This is an interesting work in concept and phenomenon. However, considering its rough materials synthesis and limited understanding of mechanism, I cannot recommend it for the publication.

1. The authors described the importance of nano-/micro-species in decreasing IR transmittance via IR reflection and IR absorption. Therefore, the type, size and content of nanomaterials should be controlled in the synthesis rather than the random pulverization of AgNWs.
2. The authors made calculations on the heat valve of the patterned ATHH hydrogels with different pattern structures. Detailed in situ characterizations should be carried out to pin down the structural evolution through the opening/closing the valve when phase transition occurred.
3. Also, there was no experimental evidence to support the influence of surface area ratio on the evaporative cooling effect.

Reviewer #2:

Remarks to the Author:

The manuscript reported an autonomous thermal homeostatic hydrogel (ATHH) that is capable of reversible and bidirectional thermal control for heat trapping at low temperatures and heat release at high temperatures. The design is interesting but there are some issues for applications. For example, hydrogels are easily dried especially for the e-skin application. So how to avoid this problem? In addition, there are some other concerns as below.

1. To be a skin-based thermoregulatory material, the temperature sensitivity, effect of moisture/humidity, mechanical durability and robustness, conformal attachment ability on human skin are important, which are not included in the manuscript.
2. How about the large scale processability of the reported hydrogels?
3. The lack of significant test between ATHH and other groups in heat trapping and release experiments (e.g., Figures 2 and 4), which makes the results unpersuasive. For example, in Figure 2c, is there a significant difference between ATHH and PNIPAm? Dried ATHH and PNIPAm on skin showed deviations of 6 °C and 3 °C, respectively. There are only 3 °C difference and the authors claimed that ATHH successfully blocked the IR. This looks not cogent.
4. The typical human-sensing temperature range is 5–50 °C (Adv. Sci. 2022, 2201525). How about the cyclic heating-cooling tests or reusage of the hydrogels at 5–60°C?
5. Line 77-80 on page 5, AgNWs should be dispersed instead of dissolved in solution. Why were the silver nanowires pulverized into micron-size particles during NIPAm polymerization involving UV irradiation?
6. In page 5, line 15, "After the polymerization of polypyrrole, transmittance is significantly reduced with an average of less than 1%, which can be attributed to the enhanced IR reflection due to additional Ag particles and IR absorption by polypyrrole". How about the other conductive polymers (e.g., polyaniline, polythiophene)? Is this a universal method?
7. For ATHH, there is a highly porous structure in the cross-sectional image (Figure 2e). Why did the PNIPAm and AgNW/PNIPAm show a densely packed structure?
8. In page 14, line 22, "All samples were dried at 90 °C for 3 h in air". Does this drying condition influence the morphology compared with freeze-drying method?

Reviewer #3:

Remarks to the Author:

Park et al describe an interesting study of thermal regulation using a hydrogel composite. Similar efforts have been documented in the literature. A recent study featuring similar ideas (based on photonics) also uses hydrogels (in that case thermochromic hydrogels) to regulate thermal transfer. Unfortunately, Park et al seem to have missed this work (ACS Photonics 2021, Zhen Fang et al., which also describes an energy autonomous system). In fact thermal homeostasis deserves attention, but documented efforts should be placed in perspective, and the basic novelty must be properly emphasized.

The bidirectional thermoregulation values 5.63 and 3.12 must be expressed and quoted with confidence error intervals, using realistic decimals.

Here a composite hydrogel (using PNIPAM) is described (ATHH), which is capable of heating and cooling (bidirectional T control). The composite is interesting.

It would be useful to consider a simple modelling based on a pore size distribution experimentally determined (histogram in Figure 2 panel i), to account for thermal scattering and heat trapping.

Is there an effect anticipated for nanometer size pores?

Unfortunately, the pore size distribution shown in Figure 2 panel i is pretty rough (the fit is very optimistic). Can this be improved?

PNIPAM has transition temperatures varying in a broad range (within 4-5 degrees). This should not be a problem for external temperatures at 10 and 50 degrees. A small complementary study could be considered varying the lower and higher temperatures in a narrower range and in finer steps, closer to LCST.

Would other LCST exhibiting polymers be suitable for different applications not related to artificial skin (e.g. construction industry/smart coatings)? Can the authors perhaps make recommendations?

The auxetic patterns applied here are very interesting. Is there an impact of the geometry/dimensions (thickness vs. lateral structural characteristics) on the thermal balance?

In conclusion, this is an interesting work, and from the perspective of this reviewer should be further considered, contingent upon addressing the above points.

REVIEWER COMMENTS

Reviewer #1 (Remarks to the Author):

In this work, the authors prepared a PNIPAM hydrogel incorporated with Ag nano-/microparticles and PPy particles, which showed the ability of bidirectional thermal regulation due to its low IR transmittance at low temperature and the accelerated heat release of porous structure at high temperature. This is an interesting work in concept and phenomenon. However, considering its rough materials synthesis and limited understanding of mechanism, I cannot recommend it for the publication.

R1-Q1. The authors described the importance of nano-/micro-species in decreasing IR transmittance via IR reflection and IR absorption. Therefore, the type, size and content of nanomaterials should be controlled in the synthesis rather than the random pulverization of AgNWs.

Answer

We appreciate the reviewer's comment. Controlling the shape type, size, and content of nano-/micro-species is of interest to many researchers. Those variables can be changed by controlling nucleation rate and nuclei growth rate through temperature regulation (J. Mater. Chem., 2000, 10, 1311±1314, Chem. Mater. 2006, 18, 6303-6307, Biotechnol. Prog. 1993, 9, 429-435), the amount and type of precursors (Adv. Mater. 1991, 9, No. 3), mechanical pulverization (Cryst. Growth Des. 2011, 11, 1742–1749) and adding surfactants (Cent. Eur. J. Chem. 6(2) • 2008 • 253–257, Turk. J. Chem. (2020) 44: 214 – 223).

In our study, shape type, size, and content control of metallic nano-/micro-species are also important because they are strongly associated with IR transmission. First, in the case of spherical silver particles, which are synthesized along with polypyrrole, it is difficult to control the size of the spherical silver particles through temperature control as the highly porous structure of polypyrrole must be formed at near-room temperature (Chem. Chem. Phys., 2004, 6, 1396). The deposition of spherical silver particles synthesized from precursors on PNIPAM surface is also difficult due to the poor adhesive force between PNIPAM and silver particle surfaces.

Therefore, we tried synthesizing silver particles using surfactants (i.e., SDBS, Tween 80 and Triton X-100) adaptable to the co-synthetic method we used. As shown in Supplementary Figure 5a, we found that Tween 80 was effective in controlling the size of the spherical silver particles, but it was difficult to synthesize particles of size near 3 μm (as depicted in Figure 2g, near 3 μm , scattering efficiency over a broad range is seen, which is preferred for effective heat trapping). Triton X-100 and SDBS show negligible effect on particle size control. (Supplementary Figure 5b and 5c). If there was a way to co-synthesize 3 μm -spherical silver particles and polypyrrole into the PNIPAM network, we could increase IR reflectance associated with the human body temperature.

The composition of polypyrrole and spherical silver particles can be controlled by the concentration of reagents. In this regard, we synthesized ATHH with various concentrations of pyrrole monomer and AgNO_3 oxidant solution. As depicted in Supplementary Figures 6a and 6b, with half or normal concentration of reagents, there are no pore blockages. However, ATHH prepared using doubled concentration of reagents used in our paper shows pore blockages, clogging the fluidic channels (Supplementary Figure 6c). As a result, as illustrated in Supplementary Figure 7, when the reagent concentrations are doubled, water evaporation rate decreases compared to samples prepared using same or lower concentration of reagents used in the paper. Thus, the concentration of reagents used in the paper shows both low IR transmission and high evaporation rate. We note that the randomly pulverized rod-shaped silver particles do not cause pore blocking, so it is suitable as an IR scattering material for integration in the highly porous ATHH network.

In regards to the rod-shaped silver particles, having a random size distribution rather than a monodispersed size is actually more effective in scattering the IR range associated with the human body temperature. As depicted in Figure 2g, rod-shaped silver particles show scattering efficiency spectra with two sharp peaks for each length. Therefore, a monodispersed particle size only covers a

narrow range of IR radiation, making it ineffective for IR scattering associated with the human body temperature.

To analyze the relation between the degree of random pulverization and IR reflectance, we measured the IR reflectance of silver nanowire (AgNW)/PNIPAm films prepared with different UV radiation time during synthesis. As shown in Supplementary Figure 2, as UV radiation time increases, more AgNWs are pulverized into rod-shaped silver particles (as demonstrated in Figure 2f); hence, the size and amount of the rod-shaped particles would vary depending on the UV radiation time. We confirmed that the optimal UV radiation time showing the highest IR reflection (i.e., highest heat trapping) was 1.6 s (Supplementary Figure 8).

To attain the optimal AgNW concentration, we measured the swelling ratio of the ATHH film at various AgNW concentrations. As shown in Supplementary Figure 9, the swelling ratio tends to decrease as the concentration of AgNW increased due to their stiffness. Since the swelling ratio is proportional to the closed/open areal ratio of auxetic-patterned ATHH, the pattern should be maximally opened (by the shrinkage of hydrogel) for efficient heat dissipation at high temperatures. However, too low a concentration of AgNW would reduce IR scattering. Therefore, we selected the ATHH film prepared with 1 wt% of AgNW, a concentration at which both an effective IR scattering and a high swelling ratio are maintained.

Added Figure and Caption in Supplementary Information

Supplementary Figure 5. Surface SEM images of ATHH films synthesized with different surfactants. ATHH films fabricated with 5 wt% of (a) Tween 80, (b) SDBS and (c) Triton X-100 during polypyrrole synthesis. Tween 80 was effective in controlling the size of the spherical silver particles, but it was difficult to synthesize particles of size near 3 μm. Triton X-100 and SDBS show negligible effect on particle size control.

Supplementary Figure 6. SEM images of freeze-dried ATHH films according to the concentration of reagents. (a) Surface and (b) cross-sectional SEM images of ATHH films prepared with half concentration of reagents and (c) cross-sectional SEM image of doubled concentration of reagents used for polypyrrole synthesizing. When ATHH are synthesized using half or normal concentration of reagents, there are no pore blockages. However, ATHH prepared using doubled concentration of reagents used in our paper shows pore blockages, clogging the fluidic channels.

Supplementary Figure 7. Evaporation rate depending on the concentration of reagents. when the concentration of the reagents for a polypyrrole synthesis is doubled, water evaporation rate decreases compared to the samples prepared using same or lower concentration of reagents used in our paper. Error bars indicate standard deviation from the mean.

Supplementary Figure 8. Reflectance spectra of AgNW/PNIPAm films with different UV radiation times during polymerization. AgNW/PNIPAm films prepared with UV radiation time of 1.6 s shows the highest IR reflectance.

Supplementary Figure 9. Swelling ratio of ATHH. ATHH films prepared with different concentration of AgNWs. V is volume of a hydrated film and V_d means volume of a fully dehydrated film. The volume of the hydrated films was measured at 20°C and 50°C for swollen and de-swollen state, respectively. Error bars indicate standard deviation from the mean.

Revised Main Text

The reduction of silver cations from the oxidant formed additional spherical silver particles at the surface and inside of the hydrogel (Supplementary Figure 4). The synthetic methods, size of particles and composition of reagents were determined by experimental analyses as shown in Supplementary Figure 5-9.

R1-Q2. The authors made calculations on the heat valve of the patterned ATHH hydrogels with different pattern structures. Detailed in situ characterizations should be carried out to pin down the structural evolution through the opening/closing the valve when phase transition occurred.

Answer

We appreciate the reviewer for the comment regarding auxetic-patterned ATHH as a heat valve. Because the LCST of ATHH is 35.7°C from the differential scanning calorimeter (DSC) data, detailed opening/closing behavior were investigated around the LCST of ATHH by pouring hot (50°C) and cold (0°C) DI water over the auxetic pattern. ATHH rapidly responded to the temperature change within few seconds. We have added real-time images in the Supplementary Information.

Added Figure and Caption in Supplementary Information

Supplementary Figure 19. Images of open and closed auxetic pattern in ATHH. a) Pouring hot water of 50°C into the ATHH in order to open the auxetic pattern and b) cold water of 0°C to close the pattern.

Revised Main Text

As depicted in Figure 3a, pattern level 3 was determined to be the optimal, with a closed/open areal ratio of 7.3, along with mechanical stability (Supplementary Figure 18). Our pattern was programmed to realize a maximum stretchability of 108% from the original state (Figure 3b), which enabled the opened area to be 70% of the total area (Figure 3c and Supplementary Video 1) above the LCST. Rapid response and temperature sensitivity of ATHH were demonstrated in Supplementary Figure 19.

R1-Q3. Also, there was no experimental evidence to support the influence of surface area ratio on the evaporative cooling effect.

Answer

We appreciate the reviewer for the comment regarding the influence of surface area ratio. As explained in Figure 3, the auxetic pattern, applied to the ATHH, opens beyond the LCST while being restricted within the confined area as shown in Supplementary Figure 19. In other words, the increase in open area caused by the shrinkage of ATHH decreases the area available for evaporation. Since the ATHH sample has a higher LCST, there is more area available for evaporation during the temperature change. Moreover, we found that the term ‘effective evaporative surface area ratio (EESAR)’ may be misunderstood as molecular surface area on the atomic scale, while our originally intended meaning had to do with the area of the hydrogel at which the evaporation of water occurs. Therefore, EESAR was changed to effective evaporative area ratio (EEAR). We have added the auxetic pattern opening sequence and the EEAR values of PNIPAm and ATHH with the increase in temperature in the Supplementary Information.

Revised Main Text

To quantitatively analyze the evaporative cooling effect, the effective evaporative area ratio (EEAR) was calculated as a function of temperature (Figure 3f). EEAR is defined as the value of evaporative area of the hydrogel after opening the auxetic pattern divided by original evaporative area before opening. EEAR gradually decreases with increasing temperature because the hydrogel gradually shrinks (Supplementary Figure 21). Since ATHH has a higher LCST, it has a larger hydrogel evaporative area over the measured temperature range compared to that of PNIPAm (i.e., the additional evaporative area is indicated by the blue colored region). Such additional surface area indicates that ATHH provides more evaporative area for water in the hydrogel to absorb heat and be evaporated, thus resulting in more effective cooling.

Reviewer #2 (Remarks to the Author):

The manuscript reported an autonomous thermal homeostatic hydrogel (ATHH) that is capable of reversible and bidirectional thermal control for heat trapping at low temperatures and heat release at high temperatures. The design is interesting but there are some issues for applications.

R2-Q1. For example, hydrogels are easily dried especially for the e-skin application. So how to avoid this problem? In addition, there are some other concerns as below.

Answer

We appreciate the reviewer for the comment regarding the dehydration of hydrogels. To overcome a dehydration problem by unwanted evaporation, Kim et al. (2020) proposed surface coating via hexane-diluted Ecoflex solution. This could delay dehydration, relieving arid hydrogel network for long-term stability. However, this type of surface coating technique hinders the cooling effect at high temperature, limiting its adaptation. Although ATHH has exposed surface area, making it prone to evaporation, it could take full advantage of its porous structure for rapid cooling under high temperature circumstances. Besides, ATHH presented outstanding restoration ability within two minutes even after being fully dehydrated.

Added Figure and Caption in Supplementary Information

Supplementary Figure 22. Hydration and dehydration states of patterned ATHH. Patterned ATHH was dried on the hot plate for one hour to fully dehydrate the hydrogel, and DI water of 24°C was poured for the hydration.

Revised Main Text

PNIPAm featured a similar evaporation rate to that of previously reported²⁸. At 45°C, the cooling power of PNIPAm was 374 W m⁻² while that of the ATHH was 487 W m⁻² (Figure 2j). As mentioned above, this can be attributed to the porosity of the ATHH. These results together further verify that ATHH effectively releases heat in the cooling state. ATHH can be rapidly restored without distortion through water supply even after being dehydrated (Supplementary Figure 22).

R2-Q2. To be a skin-based thermoregulatory material, the temperature sensitivity, effect of moisture/humidity, mechanical durability and robustness, conformal attachment ability on human skin are important, which are not included in the manuscript.

Answer

We appreciate the reviewer for the comment regarding various properties including temperature sensitivity, moisture/humidity, mechanical durability and robustness, and conformal attachment.

We measured the auxetic pattern according to temperature change by pouring hot and cold water. ATHH presented considerably rapid temperature response, immediately opening the pattern within a few seconds.

As a hydrogel composite, ATHH continuously possesses a high-level of moisture, and all experiments were conducted at a humidity of 40% (i.e., thermoregulation experiment described in Figure 4 and Figure 5). Because the exact experimental conditions were unmentioned, we added the humidity condition in the manuscript and Methods.

The mechanical durability and robustness were measured through a tensioning machine by attaching ATHH on a stretchable VHB film. By applying strain, we confirmed that ATHH could endure lateral strain until 30% elongation from the initial state. ATHH is formed only of hydrogel. Therefore, it is flexible, stretchable, and conformal to the human skin. We attached a 3M tegaderm medical tape on the wrist, and mounted patterned ATHH. Even at the bent state with lateral strain, ATHH showed stable and conformal attachment on the wrist.

Added Figure and Caption in Supplementary Information

[temperature sensitivity]

Supplementary Figure 19. Images of open and closed auxetic pattern in ATHH. a) Pouring hot water of 50°C into the ATHH in order to open the auxetic pattern and b) cold water of 0°C to close the pattern.

Revised Main Text

As depicted in Figure 3a, pattern level 3 was determined to be the optimal, with a closed/open areal ratio of 7.3, along with mechanical stability (Supplementary Figure 18). Our pattern was programmed to realize a maximum stretchability of 108% from the original state (Figure 3b), which enabled the opened area to be 70% of the total area (Figure 3c and Supplementary Video 1) above the LCST. Rapid response and temperature sensitivity of ATHH were demonstrated in Supplementary Figure 19.

[Effect of humidity and moisture]

Revised Main Text

The patterned ATHH was attached on top of a supporting substrate (made with Polyacrylamide (PAAm)/Alginate) with large openings to ensure stable mechanical motion when attached on the skin model (see Supplementary Figure 23 and methods for fabrication process, and Supplementary Figure 24 for large scale processability). These devices will be referred to as S-PNIPAm and S-ATHH hereafter, and the humidity for thermoregulation is maintained at 40%.

[mechanical durability and robustness]

Supplementary Figure 27. Mechanical durability test about stain stress of auxetic-patterned ATHH. Patterned ATHH was firmly attached on the stretchable VHB film. We fixed VHB film on the left side and applied strain for the right side by a tensioning machine.

[Conformal attachment]

Supplementary Figure 28. Conformability of auxetic-patterned ATHH on the human skin. Patterned ATHH was fixed on the wrist with a 3M medical tape to create a seamless surface. Due to the auxetic pattern and stretchable substrate made of PAAm/Alginate, ATHH conformally attached even at the bent state.

Revised Main Text

The inset of Figure 5c reveals that considerable IR released outward from the opened patterns. In the case of heating, the IR images visually show that S-ATHH skin model retain heat better than that of bare skin model (Figure 5e). After 180 s, the skin model with S-ATHH had a temperature of 22.9°C while the bare skin model had a temperature of 17.2°C (Figure 5f).

Stretchability and flexibility are crucial characteristics for skin-attachable application. Therefore, we tested mechanical strain test and conformal attachment on the skin. It was verified S-ATHH could endure up to 30% elongation. S-ATHH also presented conformal attachment even on a bent surface (Supplementary Figure 27 and 28).

R2-Q3. How about the large scale processability of the reported hydrogels?

Answer

We appreciate the reviewer for the comment regarding large-scale processability. ATHH was fabricated using lithography-patterned PDMS as a stamp, which was peeled off from a wafer mold. Therefore, it is possible to enlarge the total area of patterns, and this could be one of the strengths of ATHH.

Added Figure and Caption in Supplementary Information

Supplementary Figure 24. A photograph of S-ATHH for large area coverage. On account of the soft lithography through the silicon etching process, the size of ATHH could be regulatable.

Revised Main Text

The patterned ATHH was attached on top of a supporting substrate (made with Polyacrylamide (PAAm)/Alginate) with large openings to ensure stable mechanical motion when attached on the skin model (see Supplementary Figure 12 and methods for fabrication process, **and Supplementary Figure 24 for large-scale processability**).

R2-Q4. The lack of significant test between ATHH and other groups in heat trapping and release experiments (e.g., Figures 2 and 4), which makes the results unpersuasive. For example, in Figure 2c, is there a significant difference between ATHH and PNIPAm? Dried ATHH and PNIPAm on skin showed deviations of 6 °C and 3 °C, respectively. There are only 3 °C difference and the authors claimed that ATHH successfully blocked the IR. This looks not cogent.

Answer

We appreciate the reviewer for the comment regarding the IR blocking ability of ATHH. Thermal homeostasis has been significantly crucial for living organisms to maintain optimal state between 36°C and 37°C, adapting inner and outer temperature fluctuations. Hypothermia occurs when the core body temperature drops to 35°C or below (Annals of Medicine and Surgery, 2020, 55, 81-83), and hyperthermia starts near 40°C (International Journal of Hyperthermia, 2019, 36, 276-293). This implies that even subtle temperature deviations from thermal homeostasis incurs severe disorders and organ damage. Therefore, a temperature difference of 3°C could be significant in the aspect of thermal homeostasis.

There are textiles and materials regulating optical properties by coating or mixing other materials for heat insulation (trapping), and we made a table to compare how much temperature deviation occurs from the initial state. Omni-heat (R) is a widely used textile for heat insulation via IR blocking developed by Columbia Sportswear, and Mylar blanket, used in space, is also made of materials reflecting up to 95% of the IR wavelength range. Cai et al. (2017) demonstrated the results of both of textiles, and Omni-heat(R) and Mylar blanket showed temperature deviations of 4.4°C and 6.9°C respectively, compared to bare skin. ATHH presented temperature deviation of 6°C, which is close to the performance of Mylar. Especially in the IR wavelength range, even 3°C is substantial for maintaining thermal balance. As a comparison, we designed a heat insulation effect experiment with an aluminum film, which has an IR reflectivity of ~99% (similar with Mylar blanket), PNIPAm, and bare skin as the references.

Added Supplementary Table

Material / Textile	Optical property (R / T)	Value	Deviation temperature (from the initial, °C)	Reference
Omni-heat (R)	Reflectance	~41% (IR)	4.4°C (33°C)	16,17
Mylar blanket (space blanket)	Reflectance	~95% (IR)	6.9°C (33°C)	16
Acrylic	Reflectance	~23% (IR)	2.9°C (35°C)	18
Cotton	Reflectance	~13% (IR)	~1°C (35°C)	19, 20
ZnO/Cotton	Transmittance / Reflectance	~10% (UV) / ~45% (UV)	2.7°C ~ 4.2°C (37.1°C)	21
PNIPAm/BN-OH	Transmittance	~50% (IR)	2.5~2.9°C (29.8~30.3°C)	22
This work	Transmittance	~1% (IR)	6°C (35°C)	

Supplementary Table 2. The comparison table with other materials and textiles for heating effect. The list contains operation mechanism, value of optical properties, and temperature deviation from the initial state which were studied in previously reported papers¹⁶⁻²².

Revised Main Text

Dried ATHH and PNIPAm loaded on skin showed deviations of 6°C and 3°C, respectively, with respect to the surface temperature of the bare forearm after the saturation temperature was reached (Figure 2b and Supplementary Table 2). These results confirm that ATHH successfully blocked IR from the human body.

- 16 Cai, L., Song, A.Y., Wu, P. et al. Warming up human body by nanoporous metallized polyethylene textile. *Nat. Commun.* **8**, 496 (2017).
- 17 Ehrmann, A. & Blachowicz T. Thermal Properties of Textiles. *In Examination of Textiles Mathematical and Physical Methods*, 113-123 (Springer, 2017).
- 18 Hashan, M. M. et al. Functional properties improvement of sock items using different types of yarn. *Inter. J. Text. Sci.* **6**, 34-42 (2017).
- 19 Majumdar, A., Mukhopadhyay, S. & Yadav, R. Thermal properties of knitted fabrics made from cotton and regenerated bamboo cellulosic fibres. *Int. J. Therm. Sci.* **49**, 2042-2048 (2010).
- 20 Prakash, C. & Ramakrishnan, G. Study of thermal properties of bamboo/cotton blended single jersey knitted fabrics. *Arab. J. Sci. Eng.* **39**, 2289-2294 (2014).
- 21 Hu, R., Yang, J., Yang, P. et al. Fabrication of ZnO@Cotton fabric with anti-bacterial and radiation barrier properties using an economical and environmentally friendly method. *Cellulose* **27**, 2901–2911 (2020).
- 22 Qi, B., Wang, F., Chen, Q. et al. Enzymatic construction of a temperature-regulating fabric with multiple heat-transfer capabilities. *Cellulose* **29**, 3513–3528 (2022).

R2-Q5. The typical human-sensing temperature range is 5–50 °C (Adv. Sci. 2022, 2201525). How about the cyclic heating–cooling tests or reusability of the hydrogels at 5–50°C?

Answer

We appreciate the reviewer for the comment regarding bidirectional temperature controllability. We remeasured ATHH within the typical human-sensing temperature range from 5°C to 50°C for both S-ATHH and S-PNIPAm, and also confirmed reusability of ATHH. Due to dehydration near 50°C, we repeatedly measured temperature change in both directions (5°C to 50°C and 50°C to 5°C, respectively), with the forward and backward sweeps done separately, to observe the temperature deviation of S-ATHH. ATHH confirmed reliable and repeatable temperature deviation of $5.04^\circ\text{C} \pm 0.55^\circ\text{C}$ and $5.85^\circ\text{C} \pm 0.46^\circ\text{C}$ from the initial states at 5°C and 50°C, respectively.

Revised Figure in a manuscript

Figure 4. Autonomic thermoregulatory abilities for thermal homeostasis. a) Schematic of experimental setups for heating effect. b) Heating ability comparison, reaching target temperature until 36.5°C (top). Temperature difference from an initial state and the time consumption to reach 36.5°C (bottom). c) Schematic of experimental setups for cooling effect. d) Cooling ability comparison, placing samples on the hot stage of 40°C (top), and a minimum temperature of each samples (bottom). e) Schematic of measurement setup when the skin models were conserved at an optimum state (36.5°C), heating environmental temperature from 5°C to 50°C. f) Real-time temperature deviation measurement from 36.5°C. Dot line of graph is the environmental temperature of the thermo-hygrostat.

Added Figure and Caption in Supplementary Information

Supplementary Figure 26. Bidirectional temperature controllability of reused ATHH. The maximum temperature deviation of S-ATHH was measured at 5°C and 50°C, respectively in the thermo-hygrostat.

Revised Main Text

Moreover, an optimized auxetic pattern was designed as a heat valve to further amplify heat release at high temperatures. ATHH provided effective bidirectional thermoregulation with deviations of 5.63°C and 3.12°C $5.04^{\circ}\text{C} \pm 0.55^{\circ}\text{C}$ and $5.85^{\circ}\text{C} \pm 0.46^{\circ}\text{C}$ from the normal body temperature of 36.5°C , when the external temperatures were 5°C and 50°C , respectively.

To further confirm the bidirectional temperature controllability, S-ATHH and S-PNIPAm placed on skin models were mounted on a hot plate that maintains a temperature of 36.5°C . This set-up was then placed inside of an environmental temperature control unit. The external environmental temperature was set to gradually increase from $\pm 5^{\circ}\text{C}$ to 50°C (Figure 4e). The S-ATHH enabled the skin model to maintain a temperature closest to the homeostasis criterion at both low and high environmental temperatures (Figure 4f), thus verifying its ability for thermal homeostasis over a large temperature variation. The effect of thickness was revealed that the thicker S-ATHH reinforced heat trapping at low temperature, however, it also diminished cooling ability by interrupting heat release at high temperature (Supplementary Figure 25). The reliability and repeatability of S-ATHH was demonstrated through cyclic heating and cooling with the temperature deviation of $5.04^{\circ}\text{C} \pm 0.55^{\circ}\text{C}$ and $5.85^{\circ}\text{C} \pm 0.46^{\circ}\text{C}$ from 36.5°C , when the external temperatures were 5°C and 50°C , respectively (Supplementary Figure 26).

R2-Q6. Line 77-80 on page 5, AgNWs should be dispersed instead of dissolved in solution. Why were the silver nanowires pulverized into micron-size particles during NIPAm polymerization involving UV irradiation?

Answer

We appreciate the reviewer's comment. In the presence of O_3 , an oxide layer composed of Ag_2O is formed on AgNW by ultraviolet (UV), as stated in the paper we cited (*Sci. Rep.* **7**, 1696 (2017)). Even in the absence of O_3 , this phenomenon still occurs due to the conversion of O_2 in the atmosphere to O_3 under UV during the polymerization of AgNW/NIPAm solutions ($O_2 \rightarrow 2O, O + O_2 \rightarrow O_3$). Ag_2O is photo-sensitive and is unstable under light, so it is decomposed into Ag and AgO by UV (*Chem. Eur. J.* **2011**, *17*, 7777 – 7780); AgO is also unstable at room temperature and is further converted to Ag and O_2 . As shown in Figure 6b of (*Sci. Rep.* **7**, 1696 (2017)), AgNW becomes Ag metal debris, dispensed into the PNIPAm matrix, and oxygen escapes into the air.

Revised Figure and Caption in Supplementary Information

Supplementary Figure 2. Decomposition of AgNWs during NIPAm polymerization. When polymerization of NIPAm solution with AgNWs is proceeding, more AgNWs are pulverized by increasing duration time of UV radiation. In the presence of O_3 , an oxide layer composed of Ag_2O was formed on AgNW by ultraviolet (UV)²³. Even in the absence of O_3 , this phenomenon still occurs due to the conversion of O_2 in the atmosphere to O_3 under UV during the polymerization of AgNW/NIPAm solutions. ($O_2 \rightarrow 2O, O + O_2 \rightarrow O_3$). Ag_2O is photo-sensitive and has an unstable property in a light irradiation environment, so it is decomposed into Ag and AgO by UV, and AgO is also unstable at room temperature and is converted to Ag and O_2 ²⁴. AgNW becomes Ag metal debris, dispensed into the PNIPAm matrix, and oxygen escapes into the air. Scale bars, 5 μm .

23 Choo, D. C. & Kim, T. W. Degradation mechanisms of silver nanowire electrodes under ultraviolet irradiation and heat treatment. *Sci. Rep.* **7**, 1696 (2017).

24 Wang, X., Li, S., Yu, H., Yu, J. & Liu, S. Ag_2O as a new visible-light photocatalyst: self-stability and high photocatalytic activity. *Chemistry* **17**, 7777-7780, (2011).

R2-Q7. In page 5, line 15, “After the polymerization of polypyrrole, transmittance is significantly reduced with an average of less than 1%, which can be attributed to the enhanced IR reflection due to additional silver particles and IR absorption by polypyrrole”. How about the other conductive polymers (e.g., polyaniline, polythiophene)? Is this a universal method?

Answer

We appreciate the reviewer's comment. Polyaniline is synthesized by dissolving aniline monomer in a hydrochloric acid solution of no less than 1 M (Nanotechnology 29 (2018) 125604, Journal of Physics: Conference Series 1442 (2020) 012003) and adding an initiator. PNIPAm has an amphiphilic polymer network structure, so it is difficult to wash internal residue. We investigated materials that could be used as artificial skins and thought polyaniline was an inappropriate material due to its acidic synthesis environment. Next, Polythiophene must be synthesized in dichloromethane (DCM) as a solvent, but DCM is a toxic material and cannot be used as an artificial skin unless it is completely removed from the PNIPAm network. Therefore, we tried to synthesize polythiophene in PNIPAm through one-day aging using thiophene monomers dissolved in deionized water and FeCl_3 , but polythiophene was not synthesized, as shown in the picture below.

Revised Main Text

Methods

Finally, AgNO_3 and polypyrrole solutions were sufficiently poured over combined hydrogel, and synthesis process was kept for 2 days at the room temperature to produce fertile silver particles and polypyrrole. Other conducting polymers (i.e., polyaniline and polythiophene) are difficult to use instead of pyrrole by toxicity of residual reagent.

R2-Q8. For ATHH, there is a highly porous structure in the cross-sectional image (Figure 2e). Why did the PNIPAm and AgNW/PNIPAm show a densely packed structure?

Answer

We appreciate the reviewer's comment. In connection with R2-Q9, the reason for PNIPAm's densely packed structure in Supplementary Figure 13a is due to a shrinkage caused by thermal drying. Thermally dried AgNW/PNIPAm has densely packed structure compared to freeze-dried sample, as depicted in Supplementary Figure 13b and Supplementary Figure 3b (R2-Q9). The authors of (Phys. Chem. Chem. Phys., 2004, 6, 1396) reported significantly reduced size of PNIPAm particles from 1 μm to 400 nm when pyrrole monomers are polymerized in PNIPAm particles at 20°C, which is similar to our synthesis temperature (See Figure 1 in Phys. Chem. Chem. Phys., 2004, 6, 1396). The authors claimed that the reason for this was the attractive interaction between the hydrophobic pyrrole monomers and the hydrophobic chains of PNIPAm. To support their assumption, we observed the morphology change in the AgNW/PPy/PNIPAm synthesized using different concentrations of pyrrole monomer and AgNO₃ oxidant solutions. As depicted in Supplementary Figure 3a, when the pyrrole monomer and AgNO₃ concentrations are half the values used in our paper, the porosity is low. However, when the concentrations are doubled, the size of the composite particles forming the network decreases and the porosity increases significantly (Supplementary Figure 3b).

Added Figure and Caption in Supplementary Information

Supplementary Figure 3. Cross-sectional SEM images of the boundaries of hydrogel films. (a) ATHH synthesized using half concentration of pyrrole monomer and oxidant solutions and (b) same concentration of the reagents used in our paper. All samples were freeze-dried. when the pyrrole monomer and AgNO₃ concentrations are half the values used in our paper, the porosity is low. However, when the concentrations are doubled, the size of the composite particles forming the network decreases and the porosity increases significantly.

Revised Main Text

Next, the AgNW/PNIPAm hydrogel was immersed in a solution of pyrrole monomers and silver nitrate (used as the oxidant) for pyrrole polymerization within the hydrogel. This process generated a porous structure by significantly reduced particle size constructing hydrogel network from 400 nm to 1 μm when pyrrole monomers are polymerized in PNIPAm particles at 20°C^{35,36}. To confirm this phenomenon, we observed the morphology change in the AgNW/PPy/PNIPAm synthesized using different concentration of pyrrole monomer and AgNO₃ solutions (Supplementary Figure 3).

R2-Q9. In page 14, line 22, “All samples were dried at 90°C for 3 h in air”. Does this drying condition influence the morphology compared with freeze-drying method?

Answer

We appreciate the reviewer's comment. We investigated the effect of the drying condition on the morphology of the hydrogel. Freeze-dried PNIPAm has large pores that are not interconnected and have relatively thick outer walls that can interfere with the fluid flow (Supplementary Figure 14a). In the case of freeze-dried AgNW/PNIPAm and ATHH, they have similar porosity regardless of the drying method (Supplementary Figure 14b and 14c).

Added Figure and Caption in Supplementary Information

Supplementary Figure 14. Cross-sectional SEM images of freeze-dried hydrogel films. Freeze-dried (a) PNIPAm has the large pores that are not interconnected and have relatively thick outer walls that could interfere with the fluid flow. In the case of freeze-dried (b) AgNW/PNIPAm and (c) ATHH, they have similar porosity regardless of the drying method.

Revised Main Text

For the ATHH, a highly porous structure was observed in the cross-sectional image (Figure 2e). Unlike PNIPAm and AgNW/PNIPAm, ATHH shows a highly porous structure regardless of the drying method (Supplementary Figure 13 and 14).

Reviewer #3 (Remarks to the Author):

Park et al describe an interesting study of thermal regulation using a hydrogel composite. Similar efforts have been documented in the literature. A recent study featuring similar ideas (based on photonics) also uses hydrogels (in that case thermochromic hydrogels) to regulate thermal transfer. Unfortunately, Park et al seem to have missed this work (ACS Photonics 2021, Zhen Fang et al., which also describes an energy autonomous system).

R3-Q1. In fact, thermal homeostasis deserves attention, but documented efforts should be placed in perspective, and the basic novelty must be properly emphasized.

Answer

We appreciate the reviewer's comment. The paper (ACS Photonics 2021, Zhen Fang et al) showed a big difference from our research. First, PET and Cr/Al foil were used as the hydrogel sealing layer and the bottom layer, respectively. These materials had limited application to skin electronics due to their lack of elasticity. We conducted an analysis related to the elasticity of ATHH as demonstrated in Supplementary Figure 27 through additional experiments and added it to the revised manuscript. Second, the PNIPAm hydrogel in (ACS Photonics 2021, Zhen Fang et al) was used to filter light of wavelengths less than 2500 nm according to LCST, as shown in Figure 3H in (ACS Photonics 2021, Zhen Fang et al). However, the range of the filtered wavelengths has little effect to regulate the human body temperature. As shown in Figure 3F in (ACS Photonics 2021, Zhen Fang et al), thermal regulation was performed by introducing a Cr film which absorbs the filtered light wavelength of 2500 nm or less through PNIPAm, which too was insufficient for body temperature regulation that requires adjustment of IR wavelengths above 3 μm .

Added Figure and Caption in Supplementary Information

Supplementary Figure 27. Mechanical durability test about strain stress of auxetic-patterned ATHH. Patterned ATHH was firmly attached on the stretchable VHB film. We fixed VHB film on the left side and applied strain for the right side by a tensioning machine.

Revised Main Text

PNIPAm expels water from its hydrogel network when the environmental conditions exceed the lower critical solution temperature (LCST), which has facilitated research into diverse cooling applications such as sweating roofs²⁴, pharmaceutical packaging^{25,26}, thermal switching film²⁷ and sweating actuators²⁸.

- 27 Fang, Z. *et al.* Thermal Homeostasis Enabled by Dynamically Regulating the Passive Radiative Cooling and Solar Heating Based on a Thermochromic Hydrogel. *ACS Photonics* **8**, 2781-2790 (2021).

[mechanical durability and robustness]

The inset of Figure 5c reveals that considerable IR released outward from the opened patterns. In the case of heating, the IR images visually show that S-ATHH skin model retain heat better than that of bare skin model (Figure 5e). After 180 s, the skin model with S-ATHH had a temperature of 22.9°C while the bare skin model had a temperature of 17.2°C (Figure 5f).

Stretchability and flexibility are crucial characteristics for skin-attachable application.

Therefore, we tested mechanical strain test and conformal attachment on the skin. It was verified S-ATHH could endure up to 30% elongation. S-ATHH also presented conformal attachment even on a bent surface (Supplementary Figure 27 and 28).

R3-Q2. The bidirectional thermoregulation values 5.63 and 3.12 must be expressed and quoted with confidence error intervals, using realistic decimals.

Answer

We appreciate the reviewer's comment. We conducted bidirectional thermoregulation experiment in the typical human-sensing temperature range with 5-50°C instead of 10-50°C. Therefore, the values with error intervals were changed to $5.85^{\circ}\text{C} \pm 0.46^{\circ}\text{C}$ and $5.04^{\circ}\text{C} \pm 0.55^{\circ}\text{C}$ when the temperatures were 50°C and 5°C respectively.

Added Figure and Caption in Supplementary Information

Supplementary Figure 26. Bidirectional temperature controllability of reused ATHH. The maximum temperature deviation of S-ATHH was measured at 5°C and 50°C, respectively in the thermo-hygrostat.

Revised Main Text

Moreover, an optimized auxetic pattern was designed as a heat valve to further amplify heat release at high temperatures. ATHH provided effective bidirectional thermoregulation with deviations of 5.63°C and 3.12°C $5.04^{\circ}\text{C} \pm 0.55^{\circ}\text{C}$ and $5.85^{\circ}\text{C} \pm 0.46^{\circ}\text{C}$ from the normal body temperature of 36.5°C , when the external temperatures were 5°C and 50°C, respectively.

The S-ATHH enabled the skin model to maintain a temperature closest to the homeostasis criterion at both low and high environmental temperatures (Figure 4f), thus verifying its ability for thermal homeostasis over a large temperature variation. The effect of thickness was revealed that the thicker S-ATHH reinforced heat trapping at low temperature, however, it also diminished cooling ability by interrupting heat release at high temperature (Supplementary Figure 25). The reliability and repeatability of S-ATHH was demonstrated through cyclic heating and cooling with the deviation temperature of $5.04^{\circ}\text{C} \pm 0.55^{\circ}\text{C}$ and $5.85^{\circ}\text{C} \pm 0.46^{\circ}\text{C}$ from 36.5°C , when the external temperatures were 5°C and 50°C, respectively (Supplementary Figure 26).

R3-Q3. It would be useful to consider a simple modelling based on a pore size distribution experimentally determined (histogram in Figure 2 panel i), to account for thermal scattering and heat trapping.

Answer (Supplementary Figure 17a)

We appreciate the reviewer's comment. As the reviewer indicated, we have investigated the effect of porosity in the ATHH. To theoretically estimate the emission properties of porous PPy/PNIPAm composite related to the thermal scattering effect, optical properties (i.e., permittivity) of both PNIPAm and PPy samples were fitted using Lorentz-Drude model and FT-IR measurement data. After obtaining the permittivity spectra of both polymers, permittivity of the composite was determined using effective medium theory. Finally, the porous structure of PPy/PNIPAm composite was modelled utilizing a simple Monte-Carlo simulation by treating the pores as simple air-filled microparticles. The lower the emissivity measured, the higher the thermal scattering that occurs, which blocks thermal emission. As a result, pores of 2.6 μm diameter (most common pore size) would not decrease the emissivity of PPy/PNIPAm composite up to a porosity (i.e., f_v) of 20%.

R3-Q4. Is there an effect anticipated for nanometer size pores?

Answer (Supplementary Figure 17b)

By using the same methodology described above, we have simulated optical responses of porous composites with nanometer-sized pores. Similar to the previous result, nanometer-sized pores did not affect the emissivity of the composite film up to a porosity (f_v) of 20%.

Added Figure and Caption in Supplementary Information

Supplementary Figure 17. Spectral normal emissivity of ATHH varied with pore sizes.

Spectral normal emissivity of ATHH with pore size of (a) 2.6 μm and (b) nanometer-sized pores of 100 and 500 nm, respectively.

Supplementary Note 4. Scattering effect of porosity and pore size in ATHH

The Scattering efficiency of nano to micro-sized materials in ATHH was previously calculated by the boundary element method software (MNPBEM⁵) in Supplementary Note 2. To investigate a structural affect, We also conducted theoretical estimation of the emission characteristic of ATHH based on both porosity and pore size. First, the components of ATHH which are PNIPAm and Ppy were measured through FT-IR spectrometer with an integrating sphere. Transmittance and reflectance of components were fitted using Lorentz-Drude model⁸ as permittivity for verifying thermal scattering effect. After obtaining the permittivity spectra of components, permittivity has been determined using effective medium theory⁹. Finally porous structure of ATHH was modelled utilizing simple Monte-Carlo simulation¹⁰ by treating pores as simple air-filled microparticles. Emissivity in Supplementary Figure 17 where f_v is the porosity stands for the thermal scattering effect and heat trapping ability. High porosity in ATHH blocks an emissive heat into the outside, which causes considerable thermal scattering in the ATHH medium, and results as a low emissivity. Although emissivity is inversely proportional to the porosity, the size of the pores confirmed that it is not directly related to the value of the emissivity.

Revised Main Text

Overall, both the micro-size silver rods and spheres enhanced the reflectivity of ATHH due to their strong volumetric scattering effect (Figure 2h), thus enhancing heat trapping performance of ATHH compared with PNIPAm. We also conducted the effect of porosity and pore sizes utilizing computational simulation (Supplementary Note 4 and Supplementary Figure 17).

- 5 Waxenegger, J., Trügler, A. & Hohenester, U. Plasmonics simulations with the MNPBEM toolbox: Consideration of substrates and layer structures. *Computer Physics Communications* **193**, 138-150 (2015).
- 6 Seo, J., Qin, C., Lee, J. & Lee, B. J. Tailoring the Spectral Absorption Coefficient of a Blended Plasmonic Nanofluid Using a Customized Genetic Algorithm. *Sci. Rep.* **10**, 8891 (2020).
- 7 Rakić, A. D., Djurić, A. B., Elazar, J. M. & Majewski, M. L. Optical properties of metallic films for vertical-cavity optoelectronic devices. *Appl. Opt.* **37**, 5271-5283 (1998).
- 8 Z.M. Zhang, Nano/Microscale heat transfer, *McGraw-Hill Education* (2007).
- 9 W. Cai, *Optical Metamaterials*, Springer New York (2010).
- 10 Yalçın *et al.* Colored Radiative Cooling Coatings with Nanoparticles. *ACS Photonics* **7**, 1312-1322, (2020).

R3-Q5. Unfortunately, the pore size distribution shown in Figure 2 panel i is pretty rough (the fit is very optimistic). Can this be improved?

Answer

We appreciate the reviewer's comment. Because of the rough pore size distribution, we revised the curve type from normal to Kernel Smooth and the bandwidth as a Scott.

R3-Q6. PNIPAM has transition temperatures varying in a broad range (within 4-5 degrees). This should not be a problem for external temperatures at 10 and 50 degrees. A small complementary study could be considered varying the lower and higher temperatures in a narrower range and in finer steps, closer to LCST.

Answer

We appreciate the reviewer's comment. ATHH was fabricated from the pure PNIPAM, and the LCST of ATHH is 35.7°C from the DSC data in Supplementary Figure 1. Therefore, we observed the change of auxetic pattern near LCST by pouring hot and cold water over patterned ATHH. ATHH showed immediate response to the temperature change by shrinking and swelling, demonstrating open and closed the patterns, respectively.

Added Figure and Caption in Supplementary Information

Supplementary Figure 19. Images of open and closed auxetic pattern in ATHH pouring hot and cold water. a) Pouring hot water of 50°C into the ATHH in order to open the auxetic pattern and b) cold water of 0°C to close the pattern.

Revised Main Text

As depicted in Figure 3a, pattern level 3 was determined to be the optimal, with a closed/open areal ratio of 7.3, along with mechanical stability (Supplementary Figure 18). Our pattern was programmed to realize a maximum stretchability of 108% from the original state (Figure 3b), which enabled the opened area to be 70% of the total area (Figure 3c and Supplementary Video 1) above the LCST. Rapid response and temperature sensitivity of ATHH were demonstrated in Supplementary Figure 19.

R3-Q7. Would other LCST exhibiting polymers be suitable for different applications not related to artificial skin (e.g. construction industry/smart coatings)? Can the authors perhaps make recommendations?

Answer

We appreciate the reviewer's comment. If a hydrogel polymer has lower LCST than sea-water and a highly porous structure, we think the polymer could be used as reusable oil spill clean-up sheets. In sea-water above the LCST, hydrogel sheets would be hydrophobic and would absorb oil. The sheets can spill out the absorbed oil by being soaked in cold water under their LCST.

R3-Q8. The auxetic patterns applied here are very interesting. Is there an impact of the geometry/dimensions (thickness vs. lateral structural characteristics) on the thermal balance?

Answer

We appreciate the reviewer for the comment regarding the effect of the thickness of ATHH for thermal balance. We fabricated auxetic-patterned silicon wafer with three different thicknesses (30/60/90 μm) to make PDMS stamps. Next, the temperature of ATHH samples were measured from 5°C to 50°C. Although, the thicker ATHH samples demonstrated better heat trapping ability under the homeostasis criterion of 36.5°C, they (i.e., thicknesses of 60 and 90 μm) presented poor cooling effect over 36.5°C due to reinforced heating effect. In other words, it is better to use a thicker patterned ATHH when exposed to a lower temperature environment, and a thinner patterned ATHH when exposed to a higher temperature environment.

Added Figure and Caption in Supplementary Information

Supplementary Figure 25. The effect of S-ATHH thickness for thermal balance. The environmental temperature rose from 5°C to 50°C, and the thermal balance of three types of S-ATHH thicknesses (30 μm /60 μm /90 μm) were measured.

Revised Main Text

To further confirm the bidirectional temperature controllability, S-ATHH and S-PNIPAm placed on skin models were mounted on a hot plate that maintains a temperature of 36.5°C. This set-up was then placed inside of an environmental temperature control unit. The external environmental temperature was set to gradually increase from 5°C to 50°C (Figure 4e). The S-ATHH enabled the skin model to maintain a temperature closest to the homeostasis criterion at both low and high environmental temperatures (Figure 4f), thus verifying its ability for thermal homeostasis over a large temperature variation. The effect of thickness was revealed the thicker S-ATHH reinforced heat trapping at low temperature, however, this also diminished cooling ability by interrupting heat release at high temperature (Supplementary Figure 25). The reliability and repeatability of S-ATHH was demonstrated through cyclic heating and cooling (Supplementary Figure 26).

** See Nature Portfolio's author and referees' website at www.nature.com/authors for information about policies, services and author benefits.

This email has been sent through the Springer Nature Tracking System NY-610A-NPG&MTS

Confidentiality Statement:

This e-mail is confidential and subject to copyright. Any unauthorised use or disclosure of its contents is prohibited. If you have received this email in error please notify our Manuscript Tracking System Helpdesk team at .

Details of the confidentiality and pre-publicity policy may be found here <http://www.nature.com/authors/policies/confidentiality.html>

Privacy Policy | Update Profile

DISCLAIMER: This e-mail is confidential and should not be used by anyone who is not the original intended recipient. If you have received this e-mail in error please inform the sender and delete it from your mailbox or any other storage mechanism. Springer Nature Limited does not accept liability for any statements made which are clearly the sender's own and not expressly made on behalf of Springer Nature Ltd or one of their agents.

Please note that Springer Nature Limited and their agents and affiliates do not accept any responsibility for viruses or malware that may be contained in this e-mail or its attachments and it is your responsibility to scan the e-mail and attachments (if any).

Springer Nature Ltd. Registered office: The Campus, 4 Crinan Street, London, N1 9XW. Registered Number: 00785998 England.

Reviewers' Comments:

Reviewer #1:

Remarks to the Author:

The issues what the reviewer concerned have been addressed. I recommend it for publication.

Reviewer #3:

Remarks to the Author:

The authors provided an extensive review and gave satisfactory answers to most of this reviewer's criticism.

Regarding pore size distribution, the reviewer would suggest to add more data to improve the quality, as opposed to fitting with another formula.

The supplementary movie is nice, but it is annoying to see the film being moved and out of focus between the temperature steps. Can this be removed?

REVIEWER COMMENTS

R2-Q1. For example, hydrogels are easily dried especially for the e-skin application. So how to avoid this problem? In addition, there are some other concerns as below.

Previous answer

We appreciate the reviewer for the comment regarding the dehydration of hydrogels. To overcome a dehydration problem by unwanted evaporation, Kim et al. (2020) proposed surface coating via hexane-diluted Ecoflex solution. This could delay dehydration, relieving arid hydrogel network for long-term stability. However, this type of surface coating technique hinders the cooling effect at high temperature, limiting its adaptation. Although ATHH has exposed surface area, making it prone to evaporation, it could take full advantage of its porous structure for rapid cooling under high temperature circumstances. Besides, ATHH presented outstanding restoration ability within two minutes even after being fully dehydrated.

Figure and Caption in Supplementary Information

Supplementary Figure 22. Hydration and dehydration states of patterned ATHH. Patterned ATHH was dried on the hot plate for one hour to fully dehydrate the hydrogel, and DI water of 24°C was poured for the hydration.

Main Text

PNIPAm featured a similar evaporation rate to that of previously reported²⁸. At 45°C, the cooling power of PNIPAm was 374 W m⁻² while that of the ATHH was 487 W m⁻² (Figure 2j). As mentioned above, this can be attributed to the porosity of the ATHH. These results together further verify that ATHH effectively releases heat in the cooling state. ATHH can be rapidly restored without distortion through water supply even after being dehydrated (Supplementary Figure 22).

Revised answer

We appreciate the reviewer for the comment regarding the dehydration of hydrogels. To overcome the dehydration problem by unwanted evaporation, Kim et al. (2020) proposed surface coating via hexane-diluted Ecoflex solution. This could delay dehydration, relieving arid hydrogel network for long-term stability. However, this type of surface coating technique hinders the cooling effect at high temperature, limiting its application. Although ATHH has exposed surface area, making it prone to evaporation, it also has advantages in terms of rapid cooling at high temperatures. Furthermore, ATHH presented outstanding restoration ability within two minutes even after being fully dehydrated. We also measured the dehydration of ATHH depending on the thickness of samples at RT

(25°C) and 40°C (Supplementary Figure 22). Thus, practical time available for ATHH usage is regulatable by its thickness. Moreover, we denoted a limitation related to evaporation of the hydrogel in the discussion.

Added Figure and Caption in Supplementary Information

Supplementary Figure 24. Absolute weight change of ATHH depending on the thickness at RT and 40 °C under dehydration. We measured the weight of ATHH varying the thickness of ATHH to confirm the practical usage and reliability at each temperature.

Supplementary Figure 25. Hydration and dehydration states of patterned ATHH. Patterned ATHH was dried on the hot plate for one hour to fully dehydrate the hydrogel, and DI water of 24°C was poured for the hydration.

Revised Main Text

PNIPAm featured a similar evaporation rate to that of previously reported²⁸. At 45°C, the cooling power of PNIPAm was 374 W m^{-2} while that of the ATHH was 487 W m^{-2} (Figure 2j). As mentioned above, this can be attributed to the porosity of the ATHH. These results together further verify that ATHH effectively releases heat in the cooling state. Furthermore, practical time available for ATHH usage is regulatable by its thickness, and ATHH can be rapidly restored within 2 minutes without damage and distortion through water supply even after being dehydrated (Supplementary Figure 24 and 25).

[Discussion]

We furthermore confirmed that ATHH can deliver electrical signals that reflect changes in temperature. The dehydration of ATHH can limit its long-term usability if there is no external water supply. However, since increasing the thickness of ATHH can adjust the dehydration rate and ATHH has an outstanding restoration ability of within two minutes even after being fully dehydrated, we considered this issue resolvable through additional research. We project that ATHH will be applicable in various fields, such as treating nervous system disorders and in implementing autonomous thermoregulation in soft robotics in the near future.

R2-Q6. Line 77-80 on page 5, AgNWs should be dispersed instead of dissolved in solution. Why were the silver nanowires pulverized into micron-size particles during NIPAm polymerization involving UV irradiation?

Previous answer

We appreciate the reviewer's comment. In the presence of O_3 , an oxide layer composed of Ag_2O is formed on AgNW by ultraviolet (UV), as stated in the paper we cited (Sci. Rep. 7, 1696 (2017)). Even in the absence of O_3 , this phenomenon still occurs due to the conversion of O_2 in the atmosphere to O_3 under UV during the polymerization of AgNW/NIPAm solutions ($O_2 \rightarrow 2O$, $O + O_2 \rightarrow O_3$). Ag_2O is photo-sensitive and is unstable under light, so it is decomposed into Ag and AgO by UV (Chem. Eur. J. 2011, 17, 7777 – 7780); AgO is also unstable at room temperature and is further converted to Ag and O_2 . As shown in Figure 6b of (Sci. Rep. 7, 1696 (2017)), AgNW becomes Ag metal debris, dispensed into the PNIPAm matrix, and oxygen escapes into the air.

Figure and Caption in Supplementary Information

Supplementary Figure 2. Decomposition of AgNWs during NIPAm polymerization. When polymerization of NIPAm solution with AgNWs is proceeding, more AgNWs are pulverized by increasing duration time of UV radiation. In the presence of O_3 , an oxide layer composed of Ag_2O was formed on AgNW by ultraviolet (UV)²³. Even in the absence of O_3 , this phenomenon still occurs due to the conversion of O_2 in the atmosphere to O_3 under UV during the polymerization of AgNW/NIPAm solutions. ($O_2 \rightarrow 2O$, $O + O_2 \rightarrow O_3$). Ag_2O is photo-sensitive and has an unstable property in a light irradiation environment, so it is decomposed into Ag and AgO by UV, and AgO is also unstable at room temperature and is converted to Ag and O_2 ²⁴. AgNW becomes Ag metal debris, dispensed into the PNIPAm matrix, and oxygen escapes into the air. Scale bars, 5 μm .

23 Choo, D. C. & Kim, T. W. Degradation mechanisms of silver nanowire electrodes under ultraviolet irradiation and heat treatment. *Sci. Rep.* 7, 1696 (2017).

24 Wang, X., Li, S., Yu, H., Yu, J. & Liu, S. Ag_2O as a new visible-light photocatalyst: self-stability and high photocatalytic activity. *Chemistry* 17, 7777-7780, (2011).

Revised Answer

We conducted additional experiments to find a clearer reason why silver nanowires were pulverized into micron-size particles. Previously, we observed silver nanowire pulverization during UV curing of the AgNW/NIPAm film. This phenomenon was claimed to be caused by the formation of Ag_2O and its photodecomposition. Through additional experiments, we observed that even pristine silver nanowires were pulverized into micron-size particles according to UV exposure time. In addition, we measured sheet resistance of pristine silver nanowire films according to UV exposure time. As a result, we observed a proportional relationship between UV exposure time and sheet resistance, indicating a decrease in the electrical percolation paths due to the pulverization of silver nanowires. To support a clear understanding, we drew schematics with chemical formulas to illustrate the changes that occur in each process.

dispensed into the PNIPAm matrix. Scale bars, 5 μm (left). We also measured sheet resistance of each samples using 4-point probe method. A sample with 1.6s UV exposure time showed the highest electrical resistance due to pulverized AgNWs, while 1.0s presented relatively low resistance (right).

- 23 Choo, D. C. & Kim, T. W. Degradation mechanisms of silver nanowire electrodes under ultraviolet irradiation and heat treatment. *Sci. Rep.* **7**, 1696 (2017).
- 24 Wang, X., Li, S., Yu, H., Yu, J. & Liu, S. Ag₂O as a new visible-light photocatalyst: self-stability and high photocatalytic activity. *Chemistry* **17**, 7777-7780 (2011).
- 25 Kim, J.-H., Ma, J., Jo, S., Lee, S. & Kim, C. S. Enhancement fo antibacterial properties of a silver nanowire film via electron beam irradiation. *ACS Appl. Bio Mater.* **3**, 2117–2124 (2020).
- 26 Lin, C.-C., Lin, D.-X. & Lin, S.-H. Degradation problem in silver nanowire transparent electrodes caused by ultraviolet exposure. *Nanotechnology*, **31**, 215705 (2020).

R2-Q7. In page 5, line 15, “After the polymerization of polypyrrole, transmittance is significantly reduced with an average of less than 1%, which can be attributed to the enhanced IR reflection due to additional silver particles and IR absorption by polypyrrole”. How about the other conductive polymers (e.g., polyaniline, polythiophene)? Is this a universal method?

Previous answer

We appreciate the reviewer's comment. Polyaniline is synthesized by dissolving aniline monomer in a hydrochloric acid solution of no less than 1 M (Nanotechnology 29 (2018) 125604, Journal of Physics: Conference Series 1442 (2020) 012003) and adding an initiator. PNIPAm has an amphiphilic polymer network structure, so it is difficult to wash internal residue. We investigated materials that could be used as artificial skins and thought polyaniline was an inappropriate material due to its acidic synthesis environment. Next, Polythiophene must be synthesized in dichloromethane (DCM) as a solvent, but DCM is a toxic material and cannot be used as an artificial skin unless it is completely removed from the PNIPAm network. Therefore, we tried to synthesize polythiophene in PNIPAm through one-day aging using thiophene monomers dissolved in deionized water and FeCl_3 , but polythiophene was not synthesized, as shown in the picture below.

Revised Answer

We appreciate the reviewer's comment. It was not possible to polymerize thiophene into the PNIPAm matrix as shown in the previous answer. Instead, we tried to synthesize polyaniline (PANI) in a 0.1 M diluted hydrochloric acid solution adding ammonium persulfate. Aniline was successfully polymerized into the hydrogel matrix of ATHH. Another conducting polymer, poly(3,4-ethylenedioxythiophene) polystyrene sulfonate (PEDOT:PSS), was combined in a solution state. It was added into an already-made AgNW/NIPAm solution, and was thoroughly mixed, followed by curing through UV irradiation. We added real images of PNIPAm with PEDOT:PSS or PANI, and also measured infrared radiation (IR) transmittance. The results showed substantially low transmittance in the IR range. Embedding various conducting polymers in a PNIPAm matrix could be feasible for efficient heat conservation.

Added Figure and Caption in Supplementary Information

Supplementary Figure 11. Synthesized hydrogel films using other conductive polymers. (a) PNIPAm hydrogel films with embedded polyaniline (PANI) (left) and poly(3,4-ethylenedioxythiophene):poly(styrenesulfonate) (PEDOT:PSS) (right). (b) Measured IR transmittance of the hydrogel films in (a).

Revised methods

Finally, AgNO_3 and polypyrrole solutions were sufficiently poured over combined hydrogel, and synthesis process was kept for 2 days at the room temperature to produce fertile silver particles and polypyrrole. ~~Other conducting polymers (i.e., polyaniline and polythiophene) are difficult to use instead of pyrrole by toxicity of residual reagent.~~ Among conducting polymers, polythiophene cannot be synthesized into PNIPAm networks, and it is difficult to replace pyrrole due to the toxicity of residual reagents. However, polyaniline was readily synthesized by mixing an ammonium persulfate solution (5 wt% of NIPAm) into a solution of 10 ml of 0.1 M HCl and 1 ml ANI in 30 ml of DI water and curing for 1 day. To synthesize PEDOT:PSS embedded AgNW/PNIPAm film, 1 ml of PEDOT:PSS was mixed with already-made AgNW/PNIPAm solution. The solution was cured by UV irradiation for 1.6 seconds.

Revised Main Text

PNIPAm has a high transmittance at a wavelength range of 3–14 μm , which is associated with body heat. Thus, on its own, this material lets out most of the thermal radiation emitted by the human body, leading to considerable heat loss (Figure 2a). PNIPAm with decomposed AgNW (i.e., AgNW/PNIPAm) has reduced transmittance compared to that of the pure PNIPAm. After the polymerization of polypyrrole (ATHH), transmittance is significantly reduced with an average of less than 1%, which can be attributed to the enhanced IR reflection due to additional Ag particles and IR absorption by polypyrrole (Supplementary Figure 10). ~~Other conducting polymers could also be embedded into PNIPAm to considerably lower the transmittance (Supplementary Figure 11).~~ Such characteristics of ATHH were further verified empirically by visual inspection of images taken by an IR camera.

R2-Q8. For ATHH, there is a highly porous structure in the cross-sectional image (Figure 2e). Why did the PNIPAm and AgNW/PNIPAm show a densely packed structure?

Previous answer

We appreciate the reviewer's comment. In connection with R2-Q9, the reason for PNIPAm's densely packed structure in Supplementary Figure 13a is due to a shrinkage caused by thermal drying. Thermally dried AgNW/PNIPAm has densely packed structure compared to freeze-dried sample, as depicted in Supplementary Figure 13b and Supplementary Figure 3b (R2-Q9). The authors of (Phys. Chem. Chem. Phys., 2004, 6, 1396) reported significantly reduced size of PNIPAm particles from 1 μm to 400 nm when pyrrole monomers are polymerized in PNIPAm particles at 20°C, which is similar to our synthesis temperature (See Figure 1 in Phys. Chem. Chem. Phys., 2004, 6, 1396). The authors claimed that the reason for this was the attractive interaction between the hydrophobic pyrrole monomers and the hydrophobic chains of PNIPAm. To support their assumption, we observed the morphology change in the AgNW/PPy/PNIPAm synthesized using different concentrations of pyrrole monomer and AgNO₃ oxidant solutions. As depicted in Supplementary Figure 3a, when the pyrrole monomer and AgNO₃ concentrations are half the values used in our paper, the porosity is low. However, when the concentrations are doubled, the size of the composite particles forming the network decreases and the porosity increases significantly (Supplementary Figure 3b).

Figure and Caption in Supplementary Information

Supplementary Figure 3. Cross-sectional SEM images of the boundaries of hydrogel films. (a) ATHH synthesized using half concentration of pyrrole monomer and oxidant solutions and (b) same concentration of the reagents used in our paper. All samples were freeze-dried. when the pyrrole monomer and AgNO₃ concentrations are half the values used in our paper, the porosity is low. However, when the concentrations are doubled, the size of the composite particles forming the network decreases and the porosity increases significantly.

Main Text

Next, the AgNW/PNIPAm hydrogel was immersed in a solution of pyrrole monomers and silver nitrate (used as the oxidant) for pyrrole polymerization within the hydrogel. This process generated a porous structure by significantly reduced particle size constructing hydrogel network from 400 nm to 1 μm when pyrrole monomers are polymerized in PNIPAm particles at 20°C^{35,36}. To confirm this

phenomenon, we observed the morphology change in the AgNW/PPy/PNIPAm synthesized using different concentration of pyrrole monomer and AgNO₃ solutions (Supplementary Figure 3).

Revised Answer

We have identified the reason for the formation of the porous structure in ATHH through additional experiments. In our previous answer, we observed the cross-sectional morphology according to the concentration of polypyrrole synthesis agents (Supplementary Figure 3a). We observed that the porous structure was well-formed up to the film's boundary when the concentration of the synthesis agents was doubled. We explained this phenomenon by citing previous research that states that when polypyrrole is synthesized in PNIPAm particle at room temperature, the particle size decreases from 1 μm to 400 nm. To clarify the pore formation phenomenon further, we observed the cross-sectional morphology of ATHH by synthesizing polypyrrole for 6 and 12 hours, instead of the 48 hours used in our paper. As shown in Supplementary Figure 3b, only a small number of pores were observed in the sample synthesized for 6 hours, but more pores were formed when synthesized for 12 hours, indicating that pores are formed during the polypyrrole synthesis process. Additionally, we synthesized ATHH by increasing the concentration of silver nanowire two and four times. We observed that frames that make up the pores became denser with Ag (indicated by the increase in brightness in the SEM image) as the concentration increased, and therefore, we were able to identify that the main component that makes up the pore frame are silver nanowires (Supplementary Figure 3c). Therefore, the PNIPAm particles that are attached to the pulverized silver nanowire frame rapidly decrease in size due to polypyrrole synthesis, creating empty spaces, thus resulting in the formation of the porous structure. We included a schematic to support a clearer understanding (Supplementary Figure 3d). On the other hand, sizes of PNIPAm and AgNW/PNIPAm particles do not decrease due to the absence of polypyrrole synthesis. In addition, the PNIPAm undergo equilibrium polymeric chain conformation during air-drying due to slow water evaporation rate and becomes entangled due to Van der Waals interaction and hydrogen bonding, resulting in the densely packed structure.

Added Figure and Caption in Supplementary Information

Supplementary Figure 3. Cross-sectional SEM images of the hydrogel films. (a) Cross-sectional images of ATHH synthesized using half concentration of pyrrole monomer and oxidant solutions (left) and same concentration of the reagents used in our paper (right). When the pyrrole monomer and AgNO_3 concentrations are half the values used in our paper, the porosity is low. However, when the concentrations are doubled, the size of the composite particles forming the network decreases and the porosity increases significantly. (b) ATHH synthesized with polypyrrole for 6 hours (left) and 12 hours (right) each. The ATHH synthesized for 6 hours shows a few pores, but the sample synthesized for 12 hours has more pores within the film due to the formation of a higher amount of PPy/PNIPAm composite particles with smaller sizes. (c) ATHH synthesized with AgNW solutions of 2-fold (left) and 4-fold concentrations (right), respectively. As the concentration of AgNW increases, the frames of the pores become denser, indicating that the main component forming this frame is AgNW. All samples were freeze-dried. Scale bars, 10 μm . (d) Schematic of pore formation in ATHH during polypyrrole synthesis.

Revised Main Text

Next, the AgNW/PNIPAm hydrogel was immersed in a solution of pyrrole monomers and silver nitrate (used as the oxidant) for pyrrole polymerization within the hydrogel. This process generated

a porous structure by significantly reducing the particle size from 1 μm to 400 nm^{35,36}. To confirm this phenomenon, we observed the morphology change in the AgNW/PPy/PNIPAm synthesized using different concentration of pyrrole monomer and AgNO₃ solutions (Supplementary Figure 3). We confirmed that the porous structure was formed due to the size reduction effect of PNIPAm particles composited with pulverized AgNW frame during polypyrrole synthesis (Supplementary Figure 3).

R2-Q9. In page 14, line 22, “All samples were dried at 90°C for 3 h in air”. Does this drying condition influence the morphology compared with freeze-drying method?

Previous answer

We appreciate the reviewer's comment. We investigated the effect of the drying condition on the morphology of the hydrogel. Freeze-dried PNIPAm has large pores that are not interconnected and have relatively thick outer walls that can interfere with the fluid flow (Supplementary Figure 14a). In the case of freeze-dried AgNW/PNIPAm and ATHH, they have similar porosity regardless of the drying method (Supplementary Figure 14b and 14c).

Figure and Caption in Supplementary Information

Supplementary Figure 14. Cross-sectional SEM images of freeze-dried hydrogel films. Freeze-dried (a) PNIPAm has the large pores that are not interconnected and have relatively thick outer walls that could interfere with the fluid flow. In the case of freeze-dried (b) AgNW/PNIPAm and (c) ATHH, they have similar porosity regardless of the drying method.

Main Text

For the ATHH, a highly porous structure was observed in the cross-sectional image (Figure 2e). Unlike PNIPAm and AgNW/PNIPAm, ATHH shows a highly porous structure regardless of the drying method (Supplementary Figure 13 and 14).

Revised answer

In the previous answer in R2-Q8 and R2-Q9, we obtained a cross-sectional SEM image of freeze-dried hydrogel to compare its morphology with that of air-dried hydrogel included in the paper, and suggested that the densely packed structure in the air-dried hydrogel was due to shrinkage caused by the air-drying method in Supplementary Figure 14. We conducted additional experiments to discuss the differences in morphology according to the drying methods. When the hydrogel is freeze-dried, ice crystals are formed inside the hydrogel, and the sublimation of these ice crystals results in large pores surrounded by a thick outer wall as shown in Supplementary Figure 15a. In the case of air-drying method, the hydrogel undergoes equilibrium polymeric chain conformation due to slow water evaporation rate and becomes entangled due to Van der Waals interaction and hydrogen bonding, resulting in the generally densely packed structure [Simoni, R. C. et al. *An. Acad. Bras. Cienc.* 89, 745-755 (2017)].

When the silver nanowires are embedded in pure PNIPAm network as pore frames, they interfere with the growth of big ice crystals and the change in conformation of the hydrogel polymer, inhibiting the formation of large pores and polymer entanglement. The large pore formation by the ice crystals and the entanglement in AgNW/PNIPAm and AgNW/PPy/PNIPAm (ATHH) are also suppressed by densely dispersed silver nanowires (In ATHH, the porous structure was formed due to the size reduction effect of PNIPAm particles during polypyrrole synthesis). Therefore, they have similar morphology regardless of the drying method (Supplementary 14 and 15). We speculated that if the concentration of silver nanowires decreases, the polymer entanglement could be increased, and there may be a morphology change of ATHH depending on the drying method. To identify this hypothesis, we took cross-sectional SEM images of ATHH after air-drying and freeze-drying at reduced silver nanowire concentrations of 1/4. We identified that the air-dried sample showed densely packed structure due to the polymer entanglement (Supplementary Figure 16a), while the freeze-dried sample exhibited finely porous structure (Supplementary Figure 16b).

Revised Caption in Supplementary Information

Supplementary Figure 14. Cross-section image of air-dried PNIPAm and AgNW/PNIPAm. Unlike in the ATHH, (a) PNIPAm shows densely packed structure due to the hydrogel undergoes equilibrium polymeric chain conformation due to slow water evaporation rate and becomes entangled due to Van der Waals interaction and hydrogen bonding²⁷. (b) AgNW/PNIPAm also has a densely packed structure due to an absence of the reduction effect in the hydrogel particle size through the polypyrrole synthesis.

Revised Caption in Supplementary Information

Supplementary Figure 15. Cross-sectional SEM images of freeze-dried hydrogel films. Freeze-dried (a) PNIPAm has the large pores that are not interconnected. The formation of ice crystals inside the hydrogel and the sublimation of these ice crystals results in large pores surrounded by a thick outer wall²⁸ that could interfere with the fluid flow. In the case of freeze-dried (b) AgNW/PNIPAm and (c) ATHH, they have similar morphology regardless of the drying method due to AgNW interfere with growth of big ice crystals and the change in conformation of the hydrogel polymer.

28 Simoni, R. C. et al. Effect of drying method on mechanical, thermal and water absorption properties of enzymatically crosslinked gelatin hydrogels. *An. Acad. Bras. Cienc.* **89**, 745-755 (2017).

Added Figure and Caption in Supplementary Information

Supplementary Figure 16. Cross-sectional SEM images of air-dried and freeze-dried ATHH films with a quarter AgNW concentration. (a) Air-dried and (b) freeze-dried ATHH synthesized with AgNW solution of a quarter concentration. The air-dried sample showed densely packed structure due to the polymer entanglement, while the freeze-dried sample exhibited finely porous structure.

Revised Main Text

For the ATHH, a highly porous structure was observed in the cross-sectional image (Figure 2e). Unlike PNIPAm and AgNW/PNIPAm, ATHH shows a highly porous structure regardless of the drying method due to the reduction in the hydrogel particle size through the synthesis of polypyrrole hydrogels and an interference effect of AgNW on polymer chain conformation (Supplementary Figure 14, 15 and 16).

Reviewer #3

R3-Q1. Regarding pore size distribution, the reviewer would suggest to add more data to improve the quality, as opposed to fitting with another formula.

Answer We appreciate the reviewer for the comment regarding improving the pore size distribution figure with more data. We measured cross-sectional images of several ATHH samples, and counted precisely to confirm accurate pore size distribution. The function used in the previous figure was changed to the BiHill function in Growth/sigmoidal for non-linear fitting.

R3-Q2. The supplementary movie is nice, but it is annoying to see the film being moved and out of focus between the temperature steps. Can this be removed?

Answer

We appreciate the reviewer for comment. We tried several times to create a stable state when the hot and cold water repeatedly dropped over the auxetic patterned ATHH. Although the accumulated water was immediately pumped out, the water that was floating and trapped under the patterns hindered clear observation of the reversible response of our pattern. Therefore, we dropped water droplets far away from the pattern under observation to take a clear video. The reason behind the different responsiveness during the initial part of Supplementary Video 1 was the reduced influence of the actual temperature of the water droplet being applied to the ATHH from a distance.

Reviewers' Comments:

Reviewer #3:

Remarks to the Author:

I am satisfied with the revisions and answers provided by the authors and support publication.